# FEDERATED NEAREST NEIGHBOR MACHINE TRANSLATION

**Yichao Du**[†‡]**, Zhirui Zhang**[♮]**, Bingzhe Wu**[♮]**, Lemao Liu**[♮]**, Tong Xu**[†‡] **and Enhong Chen**[†‡]

[†]University of Science and Technology of China
[‡]State Key Laboratory of Cognitive Intelligence  [♮]Tencent AI Lab
[†‡]`duyichao@mail.ustc.edu.cn  {tongxu, cheneh}@ustc.edu.cn`
[♮]`zrustc11@gmail.com  {bingzhewu, redmondliu}@tencent.com`

## ABSTRACT

To protect user privacy and meet legal regulations, federated learning (FL) is attracting significant attention. Training neural machine translation (NMT) models with traditional FL algorithms (e.g., FedAvg) typically relies on multi-round model-based interactions. However, it is impractical and inefficient for translation tasks due to the vast communication overheads and heavy synchronization. In this paper, we propose a novel **F**ederated **N**earest **N**eighbor (FedNN) machine translation framework that, instead of multi-round model-based interactions, leverages one-round memorization-based interaction to share knowledge across different clients and build low-overhead privacy-preserving systems. The whole approach equips the public NMT model trained on large-scale accessible data with a $k$-nearest-neighbor ($k$NN) classifier and integrates the external datastore constructed by private text data from all clients to form the final FL model. A two-phase datastore encryption strategy is introduced to achieve privacy-preserving during this process. Extensive experiments show that FedNN significantly reduces computational and communication costs compared with FedAvg, while maintaining promising translation performance in different FL settings.

## 1 INTRODUCTION

In recent years, neural machine translation (NMT) has significantly improved translation quality (Bahdanau et al., 2015; Vaswani et al., 2017; Hassan et al., 2018) and has been widely adopted in many commercial systems. The current mainstream system is first built on a large-scale corpus collected by the service provider and then directly applied to translation tasks for different users and enterprises. However, this application paradigm faces two critical challenges in practice. On the one hand, previous works have shown that NMT models perform poorly in specific scenarios, especially when they are trained on the corpora from very distinct domains (Koehn & Knowles, 2017; Chu & Wang, 2018). The fine-tuning method is a popular way to mitigate the effect of domain drift, but it brings additional model deployment overhead and particularly requires high-quality in-domain data provided by users or enterprises. On the other hand, some users and enterprises pose high data security requirements due to business concerns or regulations from the government (e.g., GDPR and CCPA), meaning that we cannot directly access private data from users for model training. Thus, a conventional centralized-training manner is infeasible in these scenarios.

In response to this dilemma, a natural way is to leverage federated learning (FL) (Li et al., 2019) that enables different data owners to train a global model in a distributed manner while leaving raw private data isolated to preserve data privacy. Generally, a standard FL workflow, such as FedAvg (McMahan et al., 2017), contains multi-round model-based interactions between server and clients. At each round, the client first performs training on the local sensitive data and sends the model update to the server. The server aggregates these local updates to build an improved global model. This straightforward idea has been implemented by prior works (Roosta et al., 2021; Passban et al., 2022) that directly apply FedAvg for machine translation tasks

and introduce some parameter pruning strategies during node communication. Despite this, multi-round model-based interactions are impractical and inefficient for NMT applications. Current models heavily rely on deep neural networks as the backbone and their parameters can reach tens of millions or even hundreds of millions, bringing vast computation and communication overhead. In real-world scenarios, different clients (i.e., users and enterprises) usually have limited computation and communication capabilities, making it difficult to meet frequent model training and node communication requirements in the standard FL workflow. Further, due to the capability differences between clients, heavy synchronization also hinders the efficacy of FL workflow. Fewer interactions may ease this problem but suffer from significant performance loss.

Inspired by the recent remarkable performance of memorization-augmented techniques (e.g., the $k$-nearest-neighbor, $k$NN) in natural language processing (Khandelwal et al., 2020; 2021; Zheng et al., 2021a;b) and computer vision (Papernot & Mcdaniel, 2018; Orhan, 2018), we take a new perspective to deal with above federated NMT training problem. In this paper, we propose a novel **F**ederated **N**earest **N**eighbor (FedNN) machine translation framework, which equips the public NMT model trained on large-scale accessible data with a $k$NN classifier and integrates the external datastore constructed by private data from all clients to form the final FL model. In this way, we replace the multi-round model-based interactions in the conventional FL paradigm with the one-round encrypted memorization-based interaction to share knowledge among different clients and drastically reduce computation and communication overhead.

Specifically, FedNN follows a similar server-client architecture. The server holds large-scale accessible data to construct the public NMT model for all clients, while the client leverages their local private data to yield an external datastore that is collected to augment the public NMT model via $k$NN retrieval. Based on this architecture, the key is to merge and broadcast all datastores built from different clients, while avoiding privacy leakage. We design a two-phase datastore encryption strategy that adopts an adversarial mode between server and clients to achieve privacy-preserving during the memorization-based interaction process. On the one hand, the server builds $(\mathcal{K}, \mathcal{V})$-encryption model for clients to increase the difficulty of reconstructing the private text from the datastores constructed by other clients. The $\mathcal{K}$-encryption model is coupled with the public NMT model to ensure the correctness of $k$NN retrieval. On the other hand, all clients use a shared content-encryption model for a local datastore during the collecting process so that the server can not directly access the original datastore. During inference, the client leverages the corresponding content-decryption model to obtain the final integrated datastore.

We set up several FL scenarios (i.e., Non-IID and IID settings) with multi-domain English-German (En-De) translation dataset, and demonstrate that FedNN not only drastically decreases computation and communication costs compared with FedAvg, but also achieves the state-of-the-art translation performance in the Non-IID setting. Additional experiments verify that FedNN easily scales to large-scale clients with sparse data scenarios thanks to the memorization-based interaction across different clients. Our code is open-sourced on `https://github.com/duyichao/FedNN-MT`.

## 2 FEDNMT: FEDERATED NEURAL MACHINE TRANSLATION

Current commercial NMT systems are built on a large-scale corpus collected by the service provider and directly applied to different users and enterprises. However, this mode is difficult to flexibly satisfy the model customization and privacy protection requirements of users and enterprises. In this work, we focus on a more general application scenario, where users and enterprises participate in collaboratively training NMT models with the service provider, but the service provider cannot directly access the private data.

Formally, this application scenario consists of $|C|$ clients (i.e., user or enterprise) and a central server (i.e., service provider). The central server holds vast accessible translation data $\mathcal{D}_s = \{(\mathbf{x}_s^i, \mathbf{y}_s^i)\}_{i=1}^{|\mathcal{D}_s|}$, where $\mathbf{x}^i = (x_1^i, x_2^i, ..., x_{|\mathbf{x}^i|}^i)$ and $\mathbf{y}^i = (y_1^i, y_2^i, ..., x_{|\mathbf{y}^i|}^i)$ (for brevity, we omit the subscript $s$ here) are text sequences in the source and target languages, respectively. The central server can easily train a public NMT

model $f_\theta$ based on this corpus, where $\theta$ denotes model parameters. For each client $c$, it contains private data $\mathcal{D}_c = \{(\mathbf{x}_c^i, \mathbf{y}_c^i)\}_{i=1}^{|\mathcal{D}_c|}$, which is usually sparse in practice (i.e., $|\mathcal{D}_c| \ll |\mathcal{D}_s|$) and only accessible to itself. This setting actually falls into the federated learning framework. The straightforward idea is to apply the vanilla FL method (i.e., FedAvg) or its variants (Roosta et al., 2021; Passban et al., 2022). Generally, FedAvg contains multi-round model-based interaction updates between server and clients. At each round $r$, each client $c$ downloads a global model $f_{\theta^r}$ from the server and optimizes it using $\mathcal{D}_c$. Then the local updates $\theta_c^r$ are uploaded to the server, while the server aggregates these updates to form a new model $f_{\theta^{r+1}}$ via a simple parameter averaging technique: $\theta^{r+1} = \sum_{m=1}^{C} \frac{n_m}{n} \theta_m^r$, where $n_m$ denotes the number of data points in the $m$-th client's private data, and $n$ is the total number of all training data. However, such FL workflow is inefficient for the above application scenario because the parameter of NMT models typically reaches tens of millions or even hundreds of millions, bringing vast computation and communication overhead. The system heterogeneity between server and clients, i.e., mismatch of bandwidth, computation resources, etc., also makes it difficult to satisfy frequent updates and communication requirements in the standard FL workflow.

## 3 FEDNN: FEDERATED NEAREST NEIGHBOR MACHINE TRANSLATION

Inspired by the advanced memorization-augmented techniques, e.g., $k$NN-MT (Khandelwal et al., 2021) that has shown the promising capability of directly incorporating the pre-trained NMT model with external knowledge via $k$NN retrieval, we explore to leverage one-round memorization-based interaction rather than multi-round model-based interactions to achieve knowledge sharing across different clients. In this work, we design a novel **F**ederated **N**earest **N**eighbour (FedNN) machine translation framework, which extends the promising capability of $k$NN-MT in the federated scenario and introduces two-phase datastore encryption strategy to avoid data privacy leakage. The whole approach complements the public NMT model built by the central server with a $k$NN classifier and safely collects the local datastore constructed by private text data from all clients to form the global FL model. The entire workflow of FedNN is illustrated in Figure 1, consisting of initialization, one-round memorization-based interaction and model inference on clients.

### 3.1 INITIALIZATION

FedNN starts with the public NMT model and encryption models. The central server is responsible for optimizing the public NMT model $f_\theta$ with $\mathcal{D}_s$. Following $k$NN-MT (Khandelwal et al., 2021), the memorization (also called as datastore) is a set of key-value pairs. Given a sentence pair $(\mathbf{x}_s, \mathbf{y}_s) \in \mathcal{D}_s$, we gain the context representation $f_\theta(\mathbf{x}_s, y_{s,<t})$ in the last decoder layer at each timestep $t$. The whole datastore $\mathcal{M}_s = (\mathcal{K}_s, \mathcal{V}_s)$ is constructed by taking the representation $f_\theta(\mathbf{x}_s, y_{s,<t})$ as key and ground-truth $y_t$ as value:

$$\mathcal{M}_s = (\mathcal{K}_s, \mathcal{V}_s) = \bigcup_{(\mathbf{x}_s, \mathbf{y}_s) \in \mathcal{D}_s} \{(f_\theta(\mathbf{x}_s, y_{s,<t}), y_{s,t}), \forall y_{s,t} \in \mathbf{y}_s\}. \tag{1}$$

Based on $\mathcal{M}_s$, the central server further builds $\mathcal{K}$-Encryption model $f_{\mathcal{K}E}(.)$ that is coupled with the public NMT model. This design is for clients, which aims to increase the difficulty of reconstructing the private text from datastores constructed by other clients. The $f_{\mathcal{K}E}(.)$ should also satisfy the correctness of $k$NN retrieval during inference and the detailed $\mathcal{K}$-Encryption algorithm selection is described in Section 3.4. All clients prepare the shared content-encryption model $f_{CE}(.)$ and corresponding content-decryption model $f_{DE}(.)$, which are applied to the local datastore so that the server cannot directly access the original datastore. The content-encryption algorithm selection is relatively loose, which is detailed in Section 3.4.

### 3.2 MEMORIZATION-BASED INTERACTION

The entire memorization-based interaction is decomposed into two steps: private memorization construction and global memorization aggregation. The central server broadcasts $f_\theta$ and $f_{\mathcal{K}E}(.)$ for all clients to build the

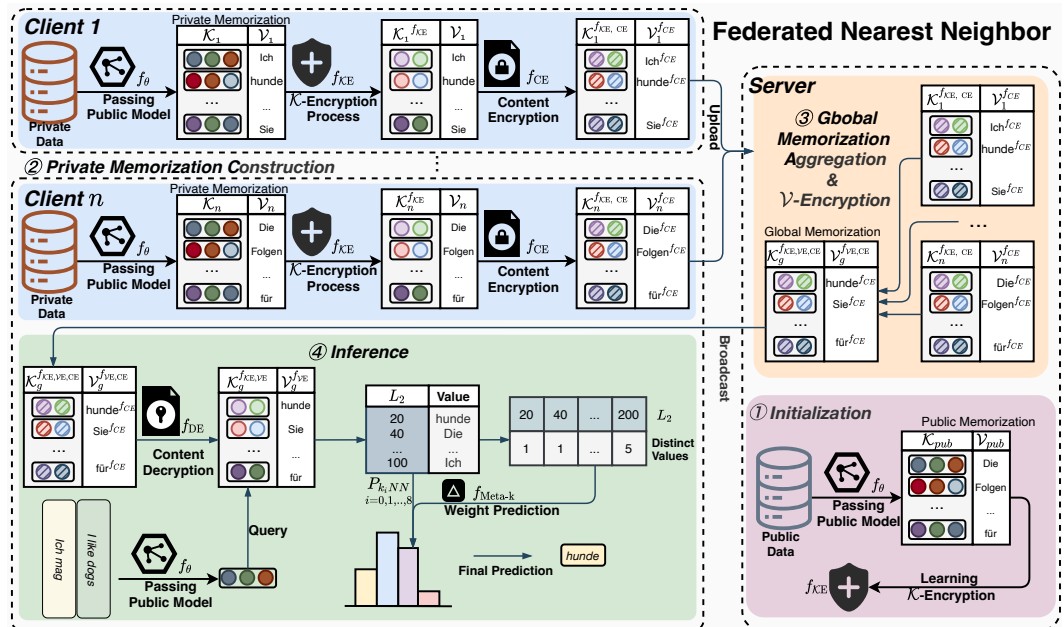

Figure 1: The overall workflow of our proposed federated framework (FedNN).

local encrypted datastore. Specifically, for each client $c$, we adopt a similar construction way as Equation 1 to yield the local datastore $\mathcal{M}_c^{f_{\mathcal{K}\text{E}}}$ via $\mathcal{D}_c$, with the difference that $f_{\mathcal{K}\text{E}}(.)$ is used to preserve private information:

$$\mathcal{M}_c^{f_{\mathcal{K}\text{E}}} = (\mathcal{K}_c^{f_{\mathcal{K}\text{E}}}, \mathcal{V}_c) = \bigcup_{(\mathbf{x}_c, \mathbf{y}_c) \in \mathcal{D}_c} \{(f_{\mathcal{K}\text{E}}(f_\theta(\mathbf{x}_c, y_{c,<t})), y_{c,t}), \forall y_{c,t} \in \mathbf{y}_c\}. \tag{2}$$

In order to ensure that above datastore is not explicitly available to the server, we encrypt key-value pairs by $f_{\text{CE}}(.)$ before uploading them to the server, formalized as:

$$\mathcal{M}_c^{f_{\mathcal{K}\text{E}}, \text{CE}} = (\mathcal{K}_c^{f_{\mathcal{K}\text{E}}, \text{CE}}, \mathcal{V}_c^{f_{\text{CE}}}) = \bigcup_{(\mathbf{x}_c, \mathbf{y}_c) \in \mathcal{D}_c} \{f_{\text{CE}}(f_{\mathcal{K}\text{E}}(f_\theta(\mathbf{x}_c, y_{c,<t})), y_{c,t}), \forall y_{c,t} \in \mathbf{y}_c\}. \tag{3}$$

Once the central server has received the private memorization from all clients, it directly aggregates all datastores via simple key-value pair concatenation and performs $\mathcal{V}$-encryption operation (i.e., shuffling on key-value pair to avoid clients identifying the source of datastore) to obtain the global memorization $\mathcal{M}_g^{f_{\mathcal{K}\text{E}}, \mathcal{V}\text{E}, \text{CE}}$, which is sent to all clients for model inference. Then each client $c$ decrypts the contents of the received $\mathcal{M}_g^{f_{\mathcal{K}\text{E}}, \mathcal{V}\text{E}, \text{CE}}$ to gain an accessible integrated datastore $\mathcal{M}_g^{f_{\mathcal{K}\text{E}}, \mathcal{V}\text{E}}$.

### 3.3 Model Inference on Clients

For model inference on clients, we follow the adaptive $k$NN-MT (AK-MT) (Zheng et al., 2021a) to incorporate $f_\theta$ with $\mathcal{M}_g^{f_{\mathcal{K}\text{E}}, \mathcal{V}\text{E}}$ via adaptive $k$NN retrieval. AK-MT introduces a lightweight Meta-$k$ Network $f_{\text{Meta-}k}$ to dynamically determine the number of retrieved tokens to consider at each step, and has promising generalization ability. Thanks to this, we could train $f_{\text{Meta-}k}$ with a small data in any vertical scenario and then directly apply it to other scenarios. Since the parameters in $f_{\text{Meta-}k}$ are negligible, we ignore the additional training and communication costs of AK-MT in FedNN.

Concretely, given the already generated words $\hat{y}_{<t}$ and source input $\mathbf{x}$, AK-MT augments the probability distribution of $t$-th target token $y_t$ via $k$NN retrieval based on the context representation $f_\theta(\mathbf{x}, \hat{y}_{<t})$. It considers a set of possible $k$s that are smaller than pre-defined $K$, i.e., $k \in \mathcal{S}$ where $\mathcal{S} = \{0\} \cup \{k_i \in \mathbb{N} \mid \log_2 k_i \in \mathbb{N}, k_i \le K\}$ ($k = 0$ indicates ignoring $k$NN retrieval and only utilizing the public NMT model). Then $K$ nearest neighbors of the current representation $f_\theta(\mathbf{x}, \hat{y}_{<t})$ are retrieved from $\mathcal{M}_g^{f_{\mathcal{K}\text{E}, \mathcal{V}\text{E}}}$ according to the squared $L_2$ distance $d(\cdot, \cdot)$. The $L_2$ distances from $f_\theta(\mathbf{x}, \hat{y}_{<t})$ to each neighbor $(h_i, v_i)$ is denoted as $d_i = d(h_i, f_\theta(\mathbf{x}, \hat{y}_{<t}))$ and the count of distinct values in top-$i$ is denoted as $c_i$. The normalized weights of applying different $k$NN retrieval results are computed as:

$$p_{\text{Meta}}(k) = \text{softmax}(f_{\text{Meta}}([d_1, ..., d_K; c_1, ..., c_K])).$$

The final prediction probability $p(y_t | \mathbf{x}, \hat{y}_{<t})$ is a weighted ensemble over differnt $k$NN retrieval distributions:

$$
\begin{aligned}
p_{k_i\text{NN}}(y_t | \mathbf{x}, \hat{y}_{<t}) &\propto \sum_{(h_i, v_i)} \mathbb{1}_{y_t = v_i} \exp\left( \frac{-d^2(h_i, f_\theta(\mathbf{x}, \hat{y}_{<t}))}{T} \right), \\
p(y_t | \mathbf{x}, \hat{y}_{<t}) &= \sum_{k_i \in \mathcal{S}} p_{\text{Meta}}(k_i) \cdot p_{k_i\text{NN}}(y_t | \mathbf{x}, \hat{y}_{<t}),
\end{aligned}
\tag{4}
$$

where $T$ is the temperature to control the sharpness of softmax function.

### 3.4 ENCRYPTION AND PRIVACY DISCUSSIONS

- $\mathcal{K}$**-Encryption.** In this work, we adopt the Product Quantizer (Jégou et al., 2011) algorithm to build $\mathcal{K}$-encryption model, which decomposes the space into a Cartesian product of low-dimensional subspaces and quantifies each subspace separately into segment code representations. Further, we map the representation to the above shortcode representation, which cannot be reversed to the original one, further reducing the possibility of reverse-constructing private data. This way also satisfies the correctness of $k$NN retrieval after encryption. Note that any algorithm that makes representation distorted and irreducible could be adopted in FedNN, such as PCA.

- **Content-Encryption.** We require to generate the ciphertext by content-encryption model and ensure that it is indistinguishable from the chosen plaintext attacks (Oded, 2004). Since each record of the memorization can be regarded as a string, it can be encrypted and decrypted using any asymmetric encryption algorithm, such as Paillier, Elgamal (Gamal, 1984) and RSA (Rivest et al., 1978), etc.

- **Threat Models and Leakage Quantifying.** We consider that all clients involved in the training process are semi-honest following prior works (Zhang et al., 2021; Bonawitz et al., 2016). In this semi-honest setting, each client adheres to the designed protocol but it may attempt to infer information about other participant's input (i.e., memorization). Under this setting, our method has achieved different protection levels for the client and server side. Specifically, for the server side, our mechanism achieves the same protection level to the conventional Public-key cryptography system (e.g., RSA). Thus, the server cannot obtain any useful information from the encrypted data. For the client side, since the client gets shared information generated by Product Quantizer (i.e., $f_{\mathcal{K}\text{E}}(f_\theta(\mathbf{x}_c, y_{c, <t})), y_{c,t})$) from other clients, the aim of this paper is to prevent the shared information from reconstruction attacks (i.e., recovering private text data from the datastore). To this end, we introduce some metrics to quantify the privacy leakage of the shared datastore information (see more details in Section 4.5).

## 4 EXPERIMENTS

### 4.1 SETUP

We adopt WMT14 En-De data (Bojar et al., 2014) and multi-domain En-De dataset (Koehn & Knowles, 2017) to simulate two typical FL scenarios for model evaluation: 1) the non-independently identically distribution

(Non-IID setting) where each client distributes data from different domains; 2) the independently identically distribution (IID setting) where each client contains the same data distribution from all domains. In our experiments, WMT14 and multi-domain En-De dataset are viewed as the server's data and clients' private data, respectively. The multi-domain data provides a natural division for exploring the Non-IID setting, of which we assign IT, Medical, and Law domain data to each of the three clients. For the IID setting, we mix the above domain data and randomly sample the same number of sentence pairs from it for each client. More dataset and implementation details can be found in Appendix A.

We compare our method **FedNN** with several baselines: (i) **Centralized Model (C-NMT)**: A standard centralized-training method uses all clients and server data to obtain a global NMT model. (ii) **Public Model (P-NMT)**: A generic NMT model is trained on only server data and used for initializing the client-side model. (iii) **FedAvg**: A vanilla FL approach (McMahan et al., 2017) that iteratively optimizes a global model through multi-round model-based interactions. (iv) **FT-Ensemble**: We use the client's private data to fine-tune P-NMT and ensemble the output probability distributions of all fine-tuned models during inference.

Table 1: BLEU score [%] of different methods on clients and server test sets. "$\triangle$" refers to the improvement of methods compared with P-NMT. The subscript "$1/\infty$" indicate that the server perform model aggregation after one or infinite epochs (i.e., client model convergence) of client model updates, respectively. The superscript "$s$" indicates that the server data is also involved in model training of FedAvg. "Comm." and "Comp." refer to communication and computational overhead in "GB" and "FLOPs" respectively.

| | Methods | Client Test | | | Server Test | Overall Performance | | | | Cost | | Inference |
| | | IT | Law | Medical | WMT14 | Client | $\triangle$ | Global | $\triangle$ | Comm. | Comp. | Speed |
|---|---|---|---|---|---|---|---|---|---|---|---|---|
| | **C-NMT** | 37.30 | 49.72 | 47.40 | 26.58 | 44.81 | — | 40.25 | — | — | — | 1.00× |
| | **P-NMT** | 26.62 | 35.91 | 30.27 | 26.63 | 30.93 | — | 29.86 | — | — | — | 1.00× |
| Non-IID | **FedAvg$_1^s$** | 26.99 | 37.65 | 32.36 | 26.63 | 32.33 | +1.40 | 30.91 | +1.05 | 388.12 | $1.72 \times 10^{19}$ | 1.00× |
| | **FedAvg$_1$** | 28.26 | 53.00 | 45.90 | 13.45 | 42.39 | +11.45 | 35.15 | +5.30 | | $3.23 \times 10^{18}$ | |
| | **FedAvg$_\infty^s$** | 27.04 | 38.37 | 32.32 | 26.33 | 32.66 | +1.72 | 31.08 | +1.22 | 4.85 | $7.02 \times 10^{17}$ | 1.00× |
| | **FedAvg$_\infty$** | 17.03 | 47.06 | 30.61 | 13.33 | 31.57 | +0.63 | 27.01 | - 2.85 | | $7.02 \times 10^{17}$ | |
| | **FT-Ensemble** | 30.11 | 38.14 | 39.15 | 17.13 | 35.80 | +4.87 | 31.13 | +1.28 | 4.85 | $7.02 \times 10^{17}$ | 0.39× |
| | **FedNN** | 35.62 | 55.57 | 49.21 | 22.29 | 46.80 | +15.87 | 40.67 | +10.82 | 5.08 | $6.72 \times 10^{15}$ | 0.75× |
| IID | **FedAvg$_1^s$** | 30.83 | 43.47 | 39.22 | 26.36 | 37.84 | +6.91 | 34.97 | +5.11 | 388.12 | $1.72 \times 10^{19}$ | 1.00× |
| | **FedAvg$_1$** | 37.99 | 54.53 | 50.23 | 14.80 | 47.58 | +16.65 | 39.39 | +9.53 | | $3.23 \times 10^{18}$ | |
| | **FedAvg$_\infty^s$** | 29.23 | 39.67 | 34.49 | 26.28 | 34.46 | +3.53 | 32.42 | +2.56 | 4.85 | $7.02 \times 10^{17}$ | 1.00× |
| | **FedAvg$_\infty$** | 34.71 | 48.68 | 44.79 | 16.23 | 42.73 | +11.79 | 36.10 | +6.25 | | $7.02 \times 10^{17}$ | |
| | **FT-Ensemble** | 36.74 | 51.34 | 47.49 | 16.42 | 45.19 | +14.26 | 38.00 | +8.14 | 4.85 | $7.02 \times 10^{17}$ | 0.39× |
| | **FedNN** | 34.64 | 54.45 | 47.98 | 23.15 | 45.69 | +14.76 | 40.06 | +10.20 | 5.08 | $6.72 \times 10^{15}$ | 0.75× |

## 4.2 MAIN RESULTS

Table 1 illustrates the performance of all methods. We observe an average 13.88 BLEU gap on the client test set between P-NMT and C-NMT. Restricted by privacy protection, it is not always possible to access private data for centralized training, so we attempt to build high-performance global models using FL techniques. Specifically, we evaluate the performance of different FL methods in both Non-IID and IID settings.

For the Non-IID setting, we have the following findings: (i) All FL methods outperform P-NMT on the client test set, but show degradation on the server test set. This indicates that FL fuses helpful information from multiple parties, but suffers from a varying degree of knowledge conflict and catastrophic forgetting. (ii)

FedAvg is heavily affected by whether the server data is used in training and model aggregation frequency. When server data is exploited during the training of FedAvg, the overall performance of FedAvg is similar to P-NMT, but FedAvg significantly improves performance on client test set without involving the server data. In addition, the performance gain has a positive correlation with the size of the client dataset (Law>Medical>IT), since FedAvg utilizes the size of different datasets as weights to aggregate client models. For the aggregation frequency, $FedAvg_1$ is much better than $FedAvg_\infty$ and more details can be found in Appendix C.2. We find that frequent aggregation significantly reduces the parameter conflicts between different models, but it brings high communication cost. (iii) FT-Ensemble is better than $FedAvg_\infty$, indicating that the fusion of output probabilities leads to less knowledge conflict compared with model aggregation. (iv) FedNN achieves an average 4.41/1.99 BLEU score improvement on the client test set compared to $FedAvg_1$ and C-NMT respectively, and maintains a competitive performance on the server test set. It demonstrates the effectiveness of FedNN in capturing client-side knowledge by memorization and integrating it with P-NMT. (v) Although FedNN slightly increases inference time, it not only improves translation quality, but also significantly reduces communication and computation overhead compared with other FL baselines, which is tolerable for clients.

For the IID setting, we have some different findings: (i) Some FL methods that do not leverage server data in their training process (i.e., $FedAvg_1$, FT-Ensemble and FedNN) outperform C-NMT on the client test set. The reason is that there is no statistical data heterogeneity among clients, resulting in fewer parameter conflicts and less conflict of probability outputs. (ii) The performance of FedNN is slightly weaker than that in the Non-IID setting. It demonstrates that the benefit of the memorization-based interaction is more significant when the data distribution is more heterogeneous.

Overall, FedNN shows stable performance with less communication and computational overhead in Non-IID and IID settings, which verifies the practicality of memorization-based interaction mechanisms. More results and analysis are shown in Appendix B.

### 4.3    THE IMPACT OF CLIENT'S NUMBER

We further verify the effectiveness of FedNN on a larger number of clients. We adopt the number of clients ranging from $(3, 6, 12, 18)$ for quick experiments.[1] The detailed results are shown in Figure 2.

- **Comparisons with FL Methods.** As the number of clients increases, we observe that: (i) Both $FedAvg_1$ and FT-Ensemble show varying degrees of performance degradation on the client test sets, especially for FT-Ensemble. We conjecture that the limited local data cannot support the training of local models and retain most of the knowledge of P-NMT. (ii) FedNN outperforms FL baselines on both private and global test sets for the Non-IID setting, while for the IID setting it maintains a similar performance to $FedAvg_1$ on private test sets and keeps a higher global performance. These results show that FedNN, benefiting from the memorization-based interaction, could quickly scale to large-scale client scenarios and avoid performance loss due to insufficient local private data. The more analysis of FL methods is described in Appendix E.
- **Comparisons with Personalized Methods.** We also compare FedNN with the personalized methods, including FT (fine-tuning P-NMT with only local client-side data) and AK-MT (constructing a datastore with only local client-side data and decoding with assisted Meta-$k$ network). AK-MT and FT perform similarly, as AK-MT is able to capture the personalized knowledge by local memorization. The performance of both AK-MT and FT tends to decrease in the Non-IID setting as the number of clients increase, while FedNN hardly decreases. For the IID setting, although the performance of all methods degrades, FedNN still achieves the best performance on all clients test sets. It is because that the global memorization capture more similar patterns than the local memorization to assist in inference.

---

[1]Due to the limited resources in our experiments, there are no more domains to ensure the Non-IID setting when the number of clients increases. Thus, we directly separate the Non-IID and IID data distributions with the ratio of $(1, \frac{1}{2}, \frac{1}{4}, \frac{1}{6})$. Note that the Non-IID setting here is not the strictly one, but it is worth exploring.

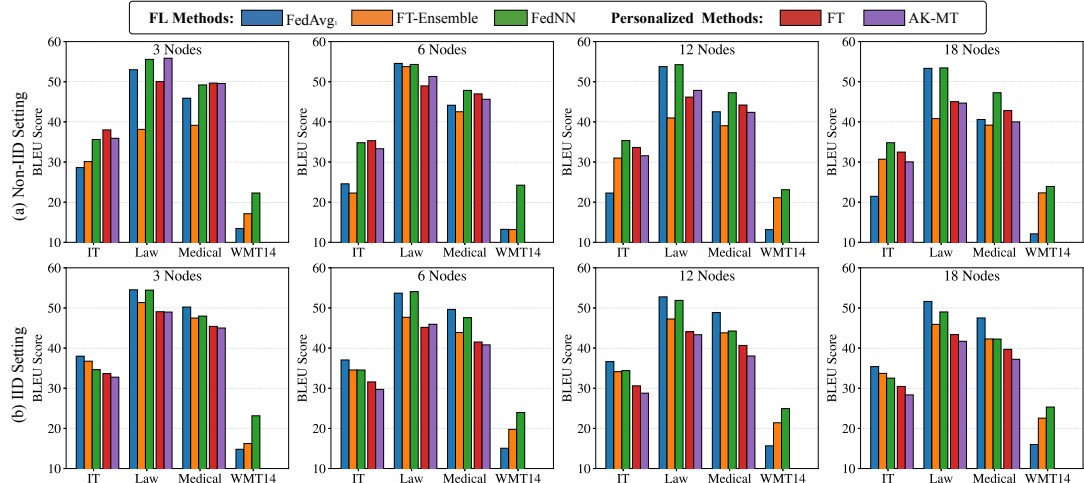

Figure 2: The translation performance of FL and personalized methods when the number of clients increases.

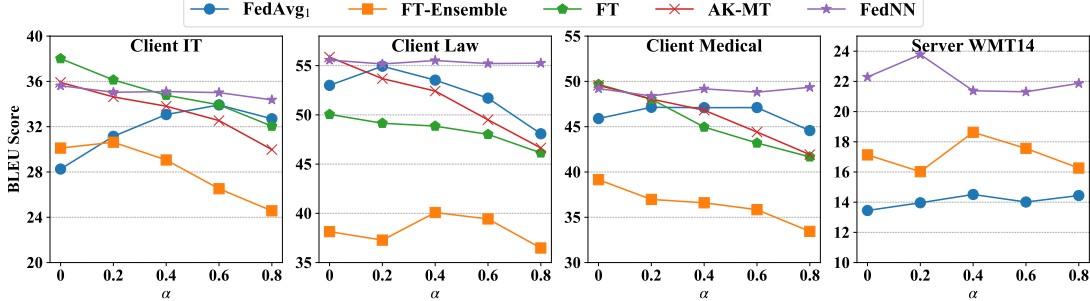

Figure 3: The impact of data distribution heterogeneity for different FL and personalized methods.

## 4.4 THE IMPACT OF DATA HETEROGENEITY

To further investigate the effect of data heterogeneity between three clients on FL performance, we adopt a mixed ratio $\alpha \in \{0, 0.2, 0.4, 0.6, 0.8\}$ to construct the data distribution that we want: we randomly take a proportion of $\alpha$ from each domain to construct the IID dataset, and then remain domain data is mixed with one-third of this IID dataset to form the final data distribution. As $\alpha \to 0$, partitions tend to be more heterogeneous (Non-IID), and conversely, the data distribution is more uniform. As shown in Figure 3, the performance of personalized methods (FT and AK-MT) is degraded as data heterogeneity decrease, which is caused by the reduction of available domain-specific data in the client. FT-Ensemble also decreases across all client test sets and is worse than FT, while FedAvg$_1$ shows opposite performance trends between Law and IT, Medical. This is because when FedAvg aggregates, the model weight of each client is proportional to the data size, and as $\alpha$ increases, the data size between clients tends from $|\mathcal{D}_{Law}| \gg |\mathcal{D}_{Medical}| \gg |\mathcal{D}_{IT}|$ to equally to $\frac{1}{3}(|\mathcal{D}_{Law}| + |\mathcal{D}_{Medical}| + |\mathcal{D}_{IT}|)$. Our FedNN maintains stable and remarkable performance across all client test sets and significantly outperforms other methods in the server test set. It indicates that the memorization-based interaction mechanism could capture and retain the knowledge of all clients, avoiding the knowledge conflict based on traditional model-based interaction.

## 4.5 QUANTITATIVE ANALYSIS OF PRIVACY

We quantify the potential privacy-leaking risks of global memorization. Since all clients obtain the public NMT model, they could utilize their own datastore to train a reverse attack model to reconstruct the private data in

global memorization. In this experiment, we task one client (e.g., IT) as the attacker and others as the defenders (e.g., Medical and Law). The reconstruction BLEU (Papineni et al., 2002)/Precision(P)/Recall(R)/F1 scores are used to evaluate the degree of privacy leakage. The more experimental details are shown in Appendix D. As illustrated in Table 2, whether the input is an unencrypted key or a key encrypted by $f_{\mathcal{K}E}$, the threat model has very low scores in all defenders, especially for the recall score, meaning that it is difficult to recover and identify valuable information from global memorization. Furthermore, the $f_{\mathcal{K}E}$ increases the difficulty of reconstructing the private text from the memorization constructed by other clients. We also provide some case studies in Appendix D.4, which better help qualitatively assess the safety of FedNN.

Table 2: The reconstruction BLEU/Precision(P)/Recall(R)/F1 score [%] of the attack model.

| Metric | Datastore | IT→Medical | IT→Law | Medical→IT | Medical→Law | Law→IT | Law→Medical |
|---|---|---|---|---|---|---|---|
| BLEU | $(\mathcal{K}, \mathcal{V})$ | 8.21 | 5.09 | 9.90 | 6.16 | 7.33 | 8.30 |
| | $(\mathcal{K}^{f_{\mathcal{K}E}}, \mathcal{V})$ | 6.52 | 4.27 | 7.88 | 5.58 | 6.35 | 6.86 |
| P/R/F1 | $(\mathcal{K}, \mathcal{V})$ | 14.55/2.90/4.84 | 35.78/7.43/12.3 | 12.18/7.53/9.31 | 23.18/11.65/15.51 | 11.15/7.04/8.63 | 9.75/4.85/6.48 |
| | $(\mathcal{K}^{f_{\mathcal{K}E}}, \mathcal{V})$ | 14.73/2.30/3.98 | 41.26/5.28/9.36 | 11.88/7.43/9.14 | 12.18/7.53/9.31 | 11.81/6.35/8.26 | 9.62/4.04/5.69 |

## 5 RELATED WORK

The FL algorithm for deep learning (McMahan et al., 2017) is first proposed for language modeling and image classification tasks. Then theory and framework of FL are widely applied to many fields, including computer vision (Lim et al., 2020), data mining (Chai et al., 2021), and edge computing (Ye et al., 2020). Recently, researchers explore applications of FL in privacy-preserving NLP, such as next word prediction (Hard et al., 2018; Chen et al., 2019), aspect sentiment classification (Qin et al., 2021), relation extraction (Sui et al., 2021), and machine translation (Roosta et al., 2021; Passban et al., 2022). For machine translation, previous works directly apply FedAvg for this task and introduce some parameter pruning strategies during node communication. However, multi-round model-based interactions are impractical and inefficient for NMT because of the huge computational and communication costs associated with large NNT models. Different from them, we design an efficient federated nearest neighbor machine translation framework that requires only one-round memorization interaction to obtain a high-quality global translation system.

Memorization-augmented methods have attracted much attention from the community and achieved remarkable performance on many NLP tasks, including language modeling (Khandelwal et al., 2020; He et al., 2021), named entity recognition (Wang et al., 2022), few-shot learning with pre-trained language model (Bari et al., 2021; Nie et al., 2022), and machine translation (Khandelwal et al., 2021; Zheng et al., 2021a;b; Wang et al., 2021; Du et al., 2022). For the NMT system, Khandelwal et al. (2021) first propose $k$NN-MT, a simple and efficient non-parametric approach that plugs $k$NN classifier over a large datastore with traditional NMT models (Vaswani et al., 2017; Zhang et al., 2018a;b; Guo et al., 2020; Wei et al., 2020) to achieve significant improvement. Our work extends the promising capability of $k$NN-MT in the federated scenario and introduces two-phase datastore encryption strategy to avoid data privacy leakage.

## 6 CONCLUSION

In this paper, we present a novel federated nearest neighbor machine translation framework to handle the federated NMT training problem. This FL framework equips the public NMT model trained on large-scale accessible data with a $k$NN classifier and safely collects all local datastores via a two-phase datastore encryption strategy to form the global FL model. Extensive experimental results demonstrate that our proposed approach significantly reduces computational and communication costs compared with FedAvg, while achieving promising performance in different FL settings. In the future, we would like to explore this approach on other sequence-to-sequence tasks. Another interesting direction is to further investigate the effectiveness of our method on a larger number of clients, such as hundreds of clients with more domains.

## ACKNOWLEDGEMENTS

We thank the anonymous reviewers for helpful feedback on early versions of this work. We appreciate Wenxiang Jiao, Xing Wang, Longyue Wang and Zhaopeng Tu for the fruitful discussions. This work was done when the first author was an intern at Tencent AI Lab and supported by the grants from National Natural Science Foundation of China (No.62222213, 62072423), and the USTC Research Funds of the Double First-Class Initiative (No.YD2150002009). Zhirui Zhang, Tong Xu and Enhong Chen are the corresponding authors.

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

## A  IMPLEMENTATION DETAILS AND EVALUATION

The statistics of the dataset and datastore used by the server/clients are listed in Table 3 and Table 4, respectively. We follow the recipe [2] to perform data pre-processing. The Moses toolkit [3] is used to tokenize all sentences and learn bpe-code in the publicly available corpus WMT14. Based on this, we split all the words of the above datasets into subword units (Sennrich et al., 2016). All experiments are implemented based on the FAIRSEQ toolkit (Ott et al., 2019). We train the public model on the WMT14 En-De dataset and use it as the initialization model for all methods. We adopt Transformer (Vaswani et al., 2017) as model structure of all baselines, in which it consists of 6 transformer encoder layers, and 6 transformer decoder layers. The input embedding size of the transformer layer is 512, the FFN layer dimension is 2048, and the number of self-attention heads is 8. During training, we deploy the Adam optimizer (Kingma & Ba, 2015) with a learning rate of 5e-4 and 4K warm-up updates to optimize model parameters. Both label smoothing coefficient and dropout rate are set to 0.1. The batch size is set to 16K tokens. We train all models with 4 Tesla-V100 GPU and set patience to 5 to select the best checkpoint on the validation set. The FAISS (Johnson et al., 2021) is leveraged to construct the datastore and we use its IndexIVFPQ strategy to implement Product Quantizer $\mathcal{K}$-encryption and fast nearest neighbor search. We utilize the FAISS to learn 4096 cluster centroids on public datastore, and apply it to client's datastore. During inference, the beam size and length penalty are set to 5 and 1 for all methods and we search 64 clusters for each target token when using FAISS. In all experiments, we report the case-sensitive BLEU score (Papineni et al., 2002) using sacreBLEU[4]. We estimate the number of floating-point operations (FLOPs) used to train the model by multiplying the training time, the number of GPUs used, and an estimation of the sustained single-precision floating-point capacity of each GPU[5].

Table 3: The statistics of datasets for server and clients.

|  | Server | Client | | |
|---|---|---|---|---|
|  | WMT14 | IT | Medical | Law |
| **Train** | 4,475,414 | 222,927 | 248,009 | 467,309 |
| **Dev** | 45,206 | 2,000 | 2,000 | 2,000 |
| **Test** | 3,003 | 2,000 | 2,000 | 2,000 |

Table 4: The statistics of datastores for server and clients.

|  | Server | Client | | | Global |
|---|---|---|---|---|---|
|  | WMT14 | IT | Medical | Law |  |
| $(\mathcal{K}, \mathcal{V})$ size | 117,427,034 | 3,085,523 | 5,858,648 | 16,868,065 | 25,812,236 |
| Hard Disk Space (Datastore) | 114 GB | 3,938 MB | 6,890 MB | 17,717 MB | 28,545 MB |
| Hard Disk Space (Faiss Index) | 8,988 MB | 244 MB | 451 MB | 1,266 MB | 1,978 MB |

---

[2]https://github.com/facebookresearch/fairseq/blob/main/examples/translation/prepare-wmt14en2de.sh

[3]https://github.com/moses-smt/mosesdecoder

[4]https://github.com/mjpost/sacrebleu, with a configuration of 13a tokenizer, case-sensitiveness, and full punctuation

[5]The single-precision floating-point capacity for Tesla-V100 GPU is 14 TFLOPs.

# B  MORE RESULTS FOR THE NON-IID SETTING

## B.1  PERFORMANCE COMPARISONS WITH CONTROLLER

We compare the performance (BLEU) and overhead of FedNN, FedAvg, and Controller in the Non-IID setting of the En2De translation task. For the Controller model, as shown in Roosta et al. (2021)'s study, 6E-6D/C-C(0-3) model achieves the best trade-off of performance and efficiency among all FL methods. Thus, we follow their setup and adopt layers 0 and 3 (both for the encoder and decoder) as controllers to participate in the parameter interaction of FedAvg training. The experimental results are shown in the Table 5. We find that the Controller has a significant performance improvement compared to P-NMT, but is still worse than $FedAvg_1$ and FedNN. In addition, since the Controller falls into the multi-round model-based FL interaction paradigm, its communication overhead is still much higher than FedNN.

Table 5: The performance and overhead comparison with *Controller.* . "Comm." and "Comp." refer to communication and computational cost in "GB" and "FLOPs" respectively

| Methods | Client Test | | | Server Test | Overall Performance | | | | Cost | |
| | IT | Law | Medical | WMT14 | Client | $\triangle$ | Global | $\triangle$ | Comm. | Comp. |
|---|---|---|---|---|---|---|---|---|---|---|
| **P-NMT** | 26.62 | 35.91 | 30.27 | 26.63 | 30.93 | – | 29.86 | – | – | – |
| **Controller** | 27.78 | 46.30 | 35.62 | 18.72 | 36.57 | +5.63 | 32.11 | +2.25 | 10.86 | $6.77 \times 10^{17}$ |
| **$FedAvg_1$** | 28.26 | 53.00 | 45.90 | 13.45 | 42.39 | +11.45 | 35.15 | +5.30 | 388.12 | $3.23 \times 10^{18}$ |
| **FedNN** | **35.62** | **55.57** | **49.21** | **22.29** | **46.80** | **+15.87** | **40.67** | **+10.82** | **5.08** | **$6.72 \times 10^{15}$** |

## B.2  EVALUATION WITH BLEURT

We evaluate the two settings in Table 1 using the neural metric - BLEURT (Sellam et al., 2020). The detailed results are shown in Table 6. We can get similar conclusions when using the BLEU score as an evaluation metric, i.e., for the Non-IID setting, our FedNN significantly outperforms all other FL methods; for the IID setting, our FedNN also achieves comparable performance to the $FedAvg_1$ and FT-Ensemble.

Table 6: BLEURT score [%] of different methods in Table 1.

| | Methods | Client Test | | | Server Test | Overall Performance | | | |
| | | IT | Law | Medical | WMT14 | Client | $\triangle$ | Global | $\triangle$ |
|---|---|---|---|---|---|---|---|---|---|
| | **C-NMT** | 70.46 | 78.49 | 74.97 | 71.62 | 74.64 | - | 73.89 | - |
| | **P-NMT** | 62.00 | 72.65 | 65.64 | 71.93 | 66.76 | - | 68.06 | - |
| **Non-IID** | **$FedAvg_1^s$** | 63.02 | 73.71 | 67.06 | 72.19 | 67.93 | +1.17 | 69.00 | +0.94 |
| | **$FedAvg_1$** | 64.09 | 78.05 | **72.74** | 53.61 | 71.63 | +4.86 | 67.12 | -0.93 |
| | **$FedAvg_\infty^s$** | 62.87 | 74.11 | 67.50 | 72.20 | 68.16 | +1.40 | 69.17 | +1.11 |
| | **$FedAvg_\infty$** | 52.63 | 77.37 | 65.02 | 56.26 | 65.01 | -1.76 | 62.82 | -5.24 |
| | **FT-Ensemble** | 64.08 | 72.47 | 70.01 | 59.17 | 68.85 | +2.09 | 66.43 | -1.62 |
| | **FedNN** | **68.86** | **78.12** | **72.74** | **67.38** | **73.24** | **+6.48** | **71.78** | **+3.72** |
| **IID** | **$FedAvg_1^s$** | 66.38 | 76.31 | 71.65 | **72.46** | 71.45 | +4.68 | **71.70** | **+3.65** |
| | **$FedAvg_1$** | 69.93 | 78.67 | **75.68** | 55.78 | **74.76** | **+8.00** | 70.02 | +1.96 |
| | **$FedAvg_\infty^s$** | 64.92 | 74.32 | 68.68 | 72.15 | 69.31 | +2.54 | 70.02 | +1.96 |
| | **$FedAvg_\infty$** | 68.96 | 78.08 | 74.39 | 59.44 | 73.81 | +7.05 | 70.22 | +2.16 |
| | **FT-Ensemble** | **70.28** | **78.80** | 75.04 | 58.78 | 74.71 | +7.94 | 70.73 | +2.67 |
| | **FedNN** | 67.69 | 77.74 | 72.31 | 68.16 | 72.58 | +5.82 | 71.48 | +3.42 |

### B.3 Significant Test for Table 1

We use the bootstrap re-sampling method to test the significant difference between FedNN and other methods. Table 7 shows the significance test results of English-German direction under the Non-IID setting. The "-" means that FedNN is not significantly better than the method. We can find that FedNN significant outperforms all FL methods, including FedAvg and FT-Ensemble.

Table 7: The significant test between FedNN and other methods for the Non-IID setting in Table 1.

| Methods | Client Test | | | Server Test |
|---|---|---|---|---|
| | IT | Law | Medical | WMT14 |
| C-NMT | - | ≤0.01 | ≤0.05 | - |
| P-NMT | ≤0.01 | ≤0.01 | ≤0.01 | ≤0.01 |
| FedAvg$_1$ | ≤0.01 | ≤0.01 | ≤0.05 | ≤0.01 |
| FT-Ensemble | ≤0.01 | ≤0.01 | ≤0.01 | ≤0.05 |

### B.4 Performance Comparisons on German-English Direction

As illustrated in Table 8, we report the performance of different FL methods in the Non-IID setting of German-English Direction. We observe that the findings in the German-English direction remain consistent with the English-German direction (shown in Table 1), in which FedNN outperforms other methods in terms of overall performance both client-side and globally.

Table 8: Performance of different methods in the German-English direction for the Non-IID setting.

| Methods | Client Test | | | Server Test | Overall Performance | | | |
|---|---|---|---|---|---|---|---|---|
| | IT | Law | Medical | WMT14 | Client | △ | Global | △ |
| P-NMT | 31.70 | 39.86 | 34.37 | 31.64 | 35.31 | - | 34.39 | - |
| FedAvg$_1$ | 32.22 | 58.32 | 48.56 | 16.83 | 46.37 | +11.06 | 38.98 | +4.59 |
| FT-Ensemble | 35.76 | 44.07 | 43.20 | 21.48 | 41.01 | +5.70 | 36.13 | +1.74 |
| FedNN | 41.11 | 60.18 | 53.44 | 27.12 | 51.58 | +16.27 | 45.46 | +11.07 |

## C Ablation Study on the Non-IID Setting

### C.1 The Impact of Client Data Size on Different FL Methods

We carry out an ablation study to verify the impact of client data size on different FL methods, including FedAvg$_1$, FT-Ensemble and FedNN. For each domain, we adopt a ratio range of $\beta \in \{0.0, 0.2, 0.4, 0.6, 0.8\}$ to randomly sample from its complete data to constitute the client data of different scales. The detailed results are shown in the Table 9. We can observe that the performance and cost of all methods increase as the size of the client data increases. Moreover, FedNN significantly outperforms other FL methods in terms of performance, communication and computational overhead.

### C.2 The Impact of Interaction Frequency on FedAvg

We conduct experiments to analyze the impact of model interaction frequency on the FedAvg performance. We set the frequency to $k \in \{1, 2, 5, 10, 20, \infty\}$, i.e., the client interacts the model with the server after $k$

Table 9: The impact of client data size on the performance and cost of different methods. "△" refers to the improvement of methods compared with P-NMT. "Comm." and "Comp." refer to communication and computational cost in "GB" and "FLOPs" respectively.

| $\beta$ | Methods | Client Test | | | Server Test | Overall Performance | | | | Cost | |
|---|---|---|---|---|---|---|---|---|---|---|---|
| | | IT | Law | Medical | WMT14 | Client | △ | Global | △ | Comm. | Comp. |
| 0.2 | **P-NMT** | 26.62 | 35.91 | 30.27 | 26.63 | 30.93 | − | 29.86 | − | − | − |
| 0.2 | **FedAvg$_1$** | 20.27 | 47.42 | 33.92 | 15.73 | 33.87 | +2.94 | 29.34 | -0.52 | 111.59 | $9.27 \times 10^{17}$ |
| | **FT-Ensemble** | **29.95** | 40.25 | 37.41 | 20.81 | 35.87 | +4.94 | 32.11 | +2.25 | 4.85 | $1.40 \times 10^{17}$ |
| | **FedNN** | 29.65 | **46.89** | **41.46** | **22.72** | **39.33** | **+8.40** | **35.18** | **+5.32** | **2.00** | **$1.34 \times 10^{15}$** |
| 0.4 | **FedAvg$_1$** | 20.99 | 48.98 | 35.58 | 16.58 | 35.18 | +4.25 | 30.53 | +0.67 | 116.44 | $9.68 \times 10^{17}$ |
| | **FT-Ensemble** | 29.60 | 39.50 | 38.45 | 21.43 | 35.85 | +4.92 | 32.25 | +2.39 | 4.85 | $2.81 \times 10^{17}$ |
| | **FedNN** | **30.41** | **50.30** | **43.89** | **23.32** | **41.53** | **+10.60** | **36.98** | **+7.12** | **2.77** | **$2.69 \times 10^{15}$** |
| 0.6 | **FedAvg$_1$** | 22.36 | 52.38 | 42.04 | 12.83 | 38.93 | +7.99 | 32.40 | +2.55 | 245.00 | $2.04 \times 10^{18}$ |
| | **FT-Ensemble** | 28.32 | 40.55 | 40.19 | 18.61 | 36.35 | +2.48 | 31.92 | +2.06 | 4.85 | $4.21 \times 10^{17}$ |
| | **FedNN** | **31.65** | **52.29** | **45.65** | **23.68** | **43.20** | **+12.26** | **38.32** | **+8.46** | **3.54** | **$4.03 \times 10^{15}$** |
| 0.8 | **FedAvg$_1$** | 24.11 | 52.71 | 43.72 | 12.76 | 40.18 | +9.25 | 33.33 | +3.47 | 354.74 | $2.92 \times 10^{17}$ |
| | **FT-Ensemble** | 29.01 | 38.16 | 39.30 | 16.87 | 35.49 | +4.56 | 30.84 | +0.98 | 4.85 | $5.62 \times 10^{17}$ |
| | **FedNN** | **32.15** | **54.11** | **46.62** | **22.22** | **44.29** | **+13.36** | **38.78** | **+8.92** | **4.31** | **$5.38 \times 10^{15}$** |

rounds of local updates. We set the total computation overhead to be the same for a fair comparison of translation performance and communication overhead. The detailed results are shown in the Table 10. We find that the performance decreases significantly as $k$ increases, especially when $k = \infty$ (i.e., a copy of the server model is trained locally until convergence), and the average performance drops to a level similar to that of P-NMT. The reason is that too many local updates suffer from catastrophic forgetting of knowledge from the previous aggregated models, resulting in strong knowledge conflicts in the new round of interactions. The optimal performance is presented at $k = 1$, which means that frequent interactions are essential to alleviate the knowledge conflicts for FedAvg.

Table 10: BLEU score [%] and communication cost of FedAvg with different interaction frequency. . "Comm. Cost" refer to communication cost in "GB".

| Methods | Client Test | | | Server Test | Overall Performance | | | | Comm. |
|---|---|---|---|---|---|---|---|---|---|
| | IT | Law | Medical | WMT14 | Client | △ | Global | △ | Cost |
| **P-NMT** | 26.62 | 35.91 | 30.27 | 26.63 | 30.93 | − | 29.86 | − | − |
| **FedAvg$_1$** | **28.26** | **53.00** | **45.90** | **13.45** | **42.39** | **+11.45** | **35.15** | **+5.30** | 388.12 |
| **FedAvg$_2$** | 26.37 | 52.92 | 44.22 | 13.07 | 41.17 | +10.24 | 34.15 | +4.29 | 194.06 |
| **FedAvg$_5$** | 24.93 | 52.23 | 41.08 | 12.85 | 39.41 | +8.48 | 32.77 | +2.92 | 77.62 |
| **FedAvg$_{10}$** | 22.63 | 51.11 | 38.60 | 12.65 | 37.45 | +6.51 | 31.25 | +1.39 | 38.81 |
| **FedAvg$_{20}$** | 21.82 | 49.53 | 36.24 | 12.75 | 35.86 | +4.93 | 30.09 | +0.23 | 19.41 |
| **FedAvg$_\infty$** | 17.30 | 47.06 | 30.61 | 13.33 | 31.57 | +0.63 | 27.01 | -2.85 | 4.85 |

## C.3 THE IMPACT OF ENSEMBLE STRATEGY ON FT-ENSEMBLE

The ensemble strategy of FT-Ensemble could be implemented in two ways: the first is to directly average the probability distribution of each client model (as used in the Table 1), i.e., FT-Ensemble; the second, similar

Table 11: BLEU score [%] of FT-Ensemble with different aggregation strategy.

| Methods | Client Test | | | Server Test | Overall Performance | | | |
|---|---|---|---|---|---|---|---|---|
| | IT | Law | Medical | WMT14 | Client | △ | Global | △ |
| **Public Model** | 26.62 | 35.91 | 30.27 | 26.63 | 30.93 | - | 29.86 | - |
| **FT-Ensemble** | 30.11 | 38.14 | 39.15 | 17.13 | 35.80 | 4.87 | 31.13 | 1.28 |
| **FT-Ensemble-Wei** | 24.09 | 48.58 | 34.11 | 16.06 | 35.59 | 4.66 | 30.71 | 0.85 |
| **FedNN** | **35.62** | **55.57** | **49.21** | **22.29** | **46.80** | **15.87** | 40.67 | 10.82 |

Table 12: The impact of P-NMT's quality. "△" refers to the improvement of the method compared with the model mentioned in Table 1.

| Methods | IT | | Law | | Medical | | WMT14 | | Clients Avg. | | Global Avg. | |
|---|---|---|---|---|---|---|---|---|---|---|---|---|
| | BLEU | △ | BLEU | △ | BLEU | △ | BLEU | △ | BLEU | △ | BLEU | △ |
| **P-NMT** | 30.72 | +4.10 | 38.69 | +2.78 | 35.90 | +5.63 | 29.77 | +3.14 | 35.10 | +4.17 | 33.77 | +3.91 |
| **FedAvg$_1$** | 28.63 | +0.37 | 58.32 | +5.32 | 49.08 | +3.18 | 16.05 | +2.60 | 45.34 | +2.95 | 38.02 | +2.87 |
| **FedNN** | 38.24 | +2.62 | 55.76 | +0.19 | 50.65 | +1.44 | 22.65 | +0.36 | 48.22 | +1.42 | 41.83 | +1.16 |

to FedAvg, is to weight the probability distribution of each client model's output by assigning weights to it according to its data size, i.e., FT-Ensemble-Wei. The performance comparison of these two ways in the Non-IID setting are shown in the Table 11. We find that FT-Ensemble outperforms FT-Ensemble-Wei in both client-side and global overall performance. FT-Ensemble has a more balanced performance on the client side, while FT-Ensemble-Wei is similar to FedAvg in that the performance is more biased towards the client Law's model, which has more local data. Our FedNN outperforms both of these methods on all clients. Note that the two implementations of FT-Ensemble described above in the IID setting are equivalent since the data size is the same for all client.

## C.4 THE IMPACT OF PUBLIC MODEL'S QUALITY

Since FedNN performs federated learning based on the P-NMT, we investigate the impact of the P-NMT's quality on performance. We introduce WMT20 En-De data to train the P-NMT, which contains 40 million parallel pairs, and conduct fast experiments in the Non-IID setting. From Table 12, we can observe that as the quality of the P-NMT improves, all methods show better performance.

## D THE DETAILS OF PRIVACY LEAKAGE ANALYSIS

### D.1 DATASET CONSTRUCTION

Given a local parallel sentence pair $(\mathbf{x}_c, \mathbf{y}_c) \in \mathcal{D}_c$ of client attacker, the public NMT model generates the context representation $k = f_\theta(\mathbf{x}_c, \mathbf{y}_{c,<t})$ in the last decoder layer at each timestep $t$, and the ground-truth is $v = y_{c,t}$. $k$ has two forms, i.e., whether it is encrypted by K-Encryption or not. Next, we concat them to obtain a training sample $r_{c,t} = k \oplus v \oplus \texttt{<2src>} \oplus \mathbf{x}_c \oplus \texttt{<2tgt>} \oplus \mathbf{y}_{c,<t} \oplus v$, where $\oplus$ is the concatenation operation. The language tag $\texttt{<2src>}$ and $\texttt{<2tgt>}$ are used to identify the generation of source and target languages, respectively. By traversing the entire $\mathcal{D}_c$, we obtain the whole dataset $\mathcal{R} = \{r_1, r_2, \ldots, r_n\}$ used to train the threat model, where $n = \sum_i^{|\mathcal{D}_c|} |\mathbf{y}_i| + |\mathcal{D}_c|$. The detailed statistics of dataset used for threat model are shown in Table 13.

Table 13: The statistics of datasets for the threat model.

|       | IT | Medical | Law |
|-------|----|---------|-----|
| **Train** | 3,085,523 | 5,858,648 | 16,868,065 |
| **Dev** | 34,737 | 55,577 | 51,423 |
| **Test** | 2,000 | 2,000 | 2,000 |

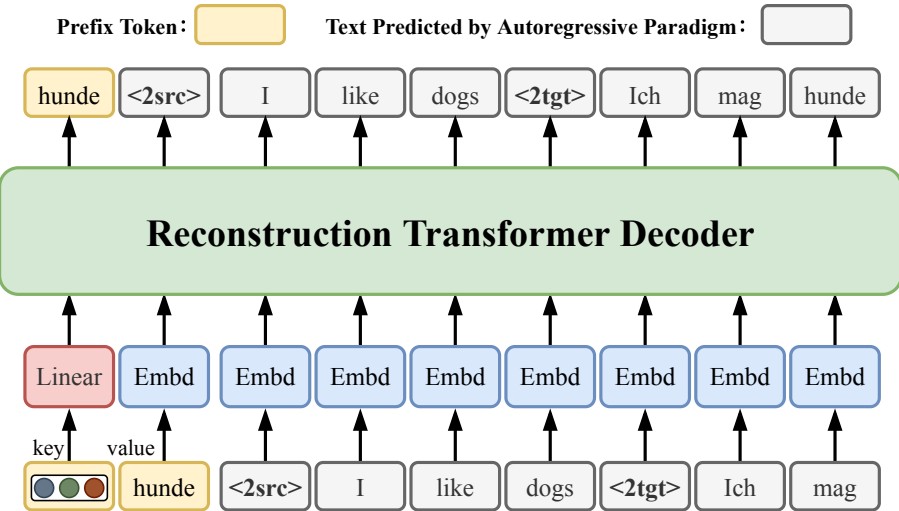

Figure 4: The threat model based on the autoregressive paradigm.

## D.2 THE ARCHITECTURE OF THREAT MODEL

The goal of the threat model is to reconstruct the corresponding original text from the memorization $(k, v)$ of client defender. As shown in Figure 4, we use a transformer decoder as the architecture of the threat model, which is similar to the left-to-right language model based on the auto-regressive paradigm. It consists of 6 transformer layers. The input embedding size is 512, the FFN layer dimension is 2048, and the number of self-attention heads is 8. We first transform the first input token $k$ to the same dimension as the word embedding using a linear layer, and then auto-regressive perform left-to-right reconstruction modeling.

## D.3 EVALUATION OF PRIVACY LEAKAGE

We quantify the privacy information leaked by global memorization using sentence-level and word-level metrics, i.e., reconstruction BLEU and privacy word hitting Precision/Recall/F1. Assuming that the text recovered by the threat model from memorization $k_i \oplus v_i$ is $\mathbf{h}_i = \{h_{i,1}, h_{i,2}, ..h_{|\mathbf{h}_i|}\}$ and the ground-truth is $\mathbf{g}_i = \{g_{i,1}, g_{i,2}, ....g_{|\mathbf{g}_i|}\}$, where $i = \{1, 2, ..., N\}$ is test sample index. Then we calculate the reconstruction BLEU score using sacreBLEU. Before evaluating the word-level privacy leakage, we require to extract the privacy dictionary of the client defender. The privacy dictionary is obtained by computing the difference between the word distribution of the defender's private dataset and the server public dataset. Further, we filter $\mathbf{h}_i$ and $\mathbf{g}_i$ according to this dictionary to obtain sentences $\mathbf{h}_i^p$ and $\mathbf{g}_i^p$ that contain only privacy words. The

word-level metric then is computed as follows:

$$\text{Precision} = \frac{\sum_i^N \sum_j^{|\mathbf{g}_i^p|} \text{Count}_{\text{hit}}(g_{i,j}^p, \mathbf{h}_i^p)}{\sum_i^N |\mathbf{h}_i^p|},$$

$$\text{Recall} = \frac{\sum_i^N \sum_j^{|\mathbf{h}_i^p|} \text{Count}_{\text{hit}}(h_{i,j}^p, \mathbf{g}_i^p)}{\sum_i^N |\mathbf{g}_i^p|}, \quad (5)$$

$$\text{F1} = \frac{2 \times \text{Precision} \times \text{Recall}}{\text{Precision} + \text{Recall}},$$

where $\text{Count}_{\text{hit}}(x, \mathbf{y})$ represents that $x$ has appeared in $\mathbf{y}$.

### D.4 QUALITATIVE ANALYSIS OF PRIVACY

Some qualitative cases are illustrated in Table 15 and we find that the style of all reconstructed texts remained consistent with the attacker's training data, including text length and domain style. For example, in Case1 and Case2, the reconstructed texts from the attack model trained on the Law domain exhibit a domain style of law client. This means that it is difficult to recover and identify valuable information, such as domain and private words, from global memorization.

## E COST ANALYSIS OF FL METHODS ON DIFFERENT CLIENT'S NUMBER

The communication and computational costs of different FL methods are illustrated in Table 14. For communication, the cost of FedAvg is much higher than that of FedNN and FT-Ensemble. The reason is that FedAvg requires multi-round communication based on the model, while both FedNN and FT-Ensemble require only one-round memorization-based communication. For computation, the cost of FT-Ensemble is linearly related to the number of nodes. It cannot be extended to practical applications because of the number of local models that need to be integrated for inference. In contrast, the cost of FedNN is only $1/60$ and $N/2$ of FedAvg$_1$ and FT-Ensemble, respectively. Considering many clients' limited communication bandwidth and computational resources, FedNN is a promising framework selection to save a lot of communication time and computational consumption.

Table 14: The communication cost and computation cost of different methods, where "M, N, R and D" respectively represent the model size (414MB), number of client, rounds of communication (160) and the total size of all encrypted datastores (1978MB).

| | | Communication Cost (GB) | | | | Computation Cost (FLOPs) | | | |
|---|---|---|---|---|---|---|---|---|---|
| | Compl. | 3 | 6 | 12 | 18 | 3 | 6 | 12 | 18 |
| **FedAvg** | M×N×R×2 | 388.12 | 776.25 | 1552.50 | 3105.00 | $3.23\times10^{18}$ | $3.23\times10^{18}$ | $3.23\times10^{18}$ | $3.23\times10^{18}$ |
| **FT-Ensemble** | M×N×(N+1) | 4.85 | 16.98 | 63.07 | 138.27 | $7.02\times10^{17}$ | $1.40\times10^{18}$ | $2.11\times10^{18}$ | $2.82\times10^{18}$ |
| **FedNN** | (D+M)×N | 5.08 | 12.08 | 26.10 | 40.12 | $6.72\times10^{15}$ | $6.72\times10^{15}$ | $6.72\times10^{15}$ | $6.72\times10^{15}$ |

## F LIMITATIONS

In this paper, we utilizes one round of memorization-based interaction to share knowledge among different clients, thus building low-overhead privacy-preserving translation systems. We discuss limitations of our method as follows.

- Despite our proposed approach achieves strong performance when exploiting global memorization sharing, it leads to reduced inference efficiency due to the need for $k$NN retrieval. As shown in Table 1, the inference

speed of FedNN is about $0.75\times$ that of P-NMT. In practice, these costs may be acceptable since we employ FAISS to speed up $k$NN retrieval. We encourage future work to improve the efficiency of $k$NN retrieval.

- The communication overhead required for memorization-based interaction is positively correlated with the client data size. Extremely large client data will make our approach inapplicable because it leads to higher communication overhead. Our approach is more applicable to the generic scenario described in Section 2, i.e., private data is sparse ($|\mathcal{D}_c| \ll |\mathcal{D}_s|$). We also encourage further exploration of how to build a smaller and more accurate memorization further to mitigate this problem.

- This paper is still very preliminary in the privacy leakage analysis of memorization interaction. Although the threat model on shared global memorization has a very low reconstruction scores, privacy leakage is still a potential risk. How to better evaluate and mitigate the privacy leakage of memorization remains an open question, which we leave for future work.

Table 15: Examples of qualitative analysis for privacy leakage. Text in green / blue represent the defender-specific ground-truth and private words, respectively. Text in red represents the hit private words by attacker. The **bold** word represents the threat model of the client-side attacker, where the superscript "$f_{\mathcal{K}E}$" represents the input of training data $k$ is encrypted $\mathcal{K}$-Encryption, otherwise it is not.

| Case Examples |
| --- |
| *Case 1: Defender is IT* |
| `<2src>` cursor `;` quickly moving `;` to an object `<2tgt>` cursor `;` schnell zu einem Objekt bewegen |
| **Medical**$^{f_{\mathcal{K}E}}$: `<2src>` curves `,` fainting, salivation, vomiting, diarrhoea, fainting, fainting, or vomiting, or diarrhoea. `<2tgt>` Kleben, Fainting, Speichelfluss, Erbrechen, Durchfall, Ohnmacht oder Erbrechen oder Durchfall oder Erbrechen |
| **Medical**: `<2src>` curonium or vecuronium: `<2tgt>` Vecuronium oder Vecuronium: |
| **Law**$^{f_{\mathcal{K}E}}$: `<2src>` palm oil falling within CN code 2710 00 90 `<2tgt>` Palmöl des KN-Codes 2710 00 90 |
| **Law**: `<2src>` curbiting the use of the designation 'butter' in Annex I to Regulation (EEC) No 3143 85 `<2tgt>` curbitration der Bezeichnung 'Butter' in Anhang I der Verordnung (EWG) Nr. 3143 / 85 |
| *Case 2: Defender is Medical* |
| `<2src>` Intravenous infusion after reconstitution and dilution. `<2tgt>` Intravenöse Infusion nach Auflösung und Verdünnung. |
| **IT**$^{f_{\mathcal{K}E}}$: `<2src>` Inserts a placeholder. `<2tgt>` Hiermit fügen Sie einen Platzhalter ein |
| **IT**: `<2src>` Inserts a new row. `<2tgt>` Fügt eine neue Zeile ein. |
| **Law**$^{f_{\mathcal{K}E}}$: `<2src>` Appointment of the date of minimum durability shall be given. `<2tgt>` Die Angabe des |
| **Law**: `<2src>` The minimum of date durability `<2tgt>` Angabe des Mindesthaltbarkeitsdatums |
| *Case 3: Defender is Law* |
| `<2src>` The Commission consistently takes a favourable view of such aid . `<2tgt>` Derartige Beihilfen werden von der Kommission stets befürwortet. |
| **IT**$^{f_{\mathcal{K}E}}$: `<2src>` The & kappname; Handbook `<2tgt>` Das Handbuch zu & kappname; |
| **IT**: `<2src>` Following packages depend on the installed packages: `<2tgt>` Die folgenden Pakete hängen von den installierten Pakete ab: |
| **Medical**$^{f_{\mathcal{K}E}}$: `<2src>` Most common side effects with Azarga (seen in between 1 and 10 patients in 100) areheadache, dizziness, somnolence (sleepiness), nausea (feeling sick), diarrhoea, abdominal tummy pain, diarrhoea, flatulence (gas), abdominal (tummy) pain, dyspepsia (indigestion), diarrhoea, nausea (feeling sick), vomiting, abdominal (tummy) pain, dyspepsia (indigestion), flatulence (wind)... |
| **Medical**: `<2src>` European Commission granted a marketing authorisation valid throughout the European Union for Nobilis Influenza H5N6 to Intervet International BV on 24 April 2009. `<2tgt>` April 2009 erteilte die Europäische Kommission dem Unternehmen Intervet International BV eine Genehmigung für das Inverkehrbringen von Nobilis Influenza H5N6 in der gesamten Europäischen Union. |

