# OpenReview forum: "Federated Nearest Neighbor Machine Translation"
_ICLR.cc/2023/Conference — ICLR 2023 poster_

### Official Review · Reviewer_61jW · 2022-10-25

**Confidence:** 4
**Correctness:** 4
**Technical Novelty And Significance:** 2
**Empirical Novelty And Significance:** 4
**Recommendation:** 6

**Clarity, Quality, Novelty And Reproducibility:**

These points are all written in the above Strength and Weakness text box.
Please refer it.


**Details Of Ethics Concerns:**

I found no ethical concerns.

**Strength And Weaknesses:**

Strength:

* This paper is well-organized and easy to read.
* Figure 1 is neat and concise, and can help a deeper understanding of the proposed method.
* The claims and the proposed method seem reasonable.
* The experiments show that the proposed method can perform better than a few conventional methods in practical federated learning scenarios.

Weakness:

* The primary idea of the proposed method is just borrowing the previous method of kNN-MT. From this perspective, the proposed method can be considered just applying the existing method (with slight modifications) in a different learning configuration.
* most of the sub-modules that consist of the proposed method are also a combination of the existing methods. Therefore, the method is not innovative, so this is somewhat of an incremental study.

**Summary Of The Paper:**

This paper tackles developing neural machine translation models in the federated learning scenario.
There have been several methods that attempt to solve the identical scenario.
The authors point out that the conventional methods require vast communication overheads and heavy synchronization that makes the methods impractical when we consider the actual use-case for training machine translation models, including the way to use training data with privacy.
This paper proposed a method that can be built only with low overhead while preserving privacy to mitigate inefficiency.
The key idea is to leverage the technique of nearest-neighbor neural machine translation (kNN-MT).
The experimental results on two datasets in the federated learning scenario show significantly better performance over several conventional methods, with fewer communication costs.

**Summary Of The Review:**

As I pointed out above, the method itself is not super novel, but this paper carefully considers and tackles a practical scenario in actual use.
Therefore, this paper has sufficient contributions to be accepted to the conference.

---

> ### Author Response · Authors · 2022-11-19
> **Response to Reviewer 61jW**
>
> We would like to thank for your time and all nice suggestions towards our paper! We will polish the expressions and revise the paper carefully. According to your comments, we provide the responses as follows:
>
> > **Q1:  The novelty of the proposed method is incremental.**
>
> Please refer to Q1 of General Response.

---

> ### Author Response · Authors · 2022-12-09
> **Re: Thanks again!**
>
> We thank the reviewer for the review. We have provided clarification in the author's response. Should the reviewer have any further suggestions, we would be happy to provide further clarification and revise our manuscript.

---

### Official Review · Reviewer_acCF · 2022-10-25

**Confidence:** 4
**Correctness:** 4
**Technical Novelty And Significance:** 2
**Empirical Novelty And Significance:** 3
**Recommendation:** 6

**Clarity, Quality, Novelty And Reproducibility:**

Clarity & Quality: the paper is well-written and easy to follow. There are a few places that I am a bit confused about and I would suggest the authors make them more clearly or include more discussions.
- In Table 1, there is too much content. It is better to highlight some numbers.
- In Table 1, why is the communication/computation of FedAvg_1 more than FedAvg_inf?
- In Table 1, the centralized model shows slightly worse performance compared to FedNN. This is surprising to me, as the centralized model is trained with client data. Could you provide more discussion about this?

Novelty: See my evaluation in the previous section.

Reproducibility: I believe the results can be reproduced easily. I encourage authors to include more implementation details.


**Strength And Weaknesses:**

Strengths:
- The paper studies an intersection of two important topics -- machine translation and federated learning. I believe this topic can be potentially impactful.
- The paper is well-motivated and the general idea of building domain-specific datastores without updating the model in the federated learning setting makes sense to me.
- The paper is well-written and easy to follow.

Weaknesses:
- The main concern of this paper is that it may not make enough technical contributions. The FedNN framework is an extension of the kNN-MT in the federated learning setup, but only this contribution doesn’t guarantee the paper to be published in ICLR in my opinion. The two-phase datastore encryption is interesting, however it is not studied extensively in the paper but built on off-the-shelf tools. The paper can be made stronger by considering different encryption methods.
- The paper compares the performance and communication/computational overhead of FedNN and existing methods. However, the paper does not discuss the privacy preserving ability of different approaches. For example, how do multi-round model-based interaction methods preserve the privacy of clients compared to FedNN?


**Summary Of The Paper:**

The paper proposes the federated nearest neighbor (FedNN) machine translation framework, which is built on top of the nearest neighbor machine translation method (kNN-MT). FedNN extends kNN-MT in a federated learning setting, where a server provides a global model and clients keep private data to improve the model. The paper proposes that to avoid privacy leakage, the clients send out an encrypted datastore which is built on a private dataset. By doing this, the system avoids multi-round model-based interactions, so that the communication overhead is made less. The paper conducts experiments on WMT14 and multi-domain en-de dataset. The experiments show that FedNN outperforms several baselines while keeping the communication and computational overhead low.

The main contribution of this paper is to apply kNN-MT to a federated learning setup, which is a reasonable application given kNN-MT is a cheap and effective method for domain adaptation.


**Summary Of The Review:**

In summary, I think the paper studies an important problem and the proposed solution makes sense to me. However, I also think the paper can still be improved in some places.

---

> ### Author Response · Authors · 2022-11-19
> **Response to Reviewer acCF**
>
> We would like to thank for your time and all nice suggestions towards our paper! We will polish the expressions and revise the paper carefully. According to your comments, we provide the responses as follows:
>
> > **Q1: May not make enough technical contributions.**
>
> Please refer to Q1 of General Response.
>
> >  **Q2: Why is the communication/computation of FedAvg$_1$ more than FedAvg$_{\infty}$?**
>
> Since there are serious knowledge conflicts between different clients' models, FedAvg$_1$ requires huge rounds of iterative interactions to ease this issue. In contrast, in FedAvg$_\infty$, each client only fine-tunes a copy of the server model to convergence locally and then uploads it to the server for aggregation to complete the entire FL workflow. Clearly, the communication/computational cost of performing multiple rounds of interactions is significantly higher for FedAvg$_1$ than one round of interactions for FedAvg$_\infty$.
>
> > **Q3: The two-phase datastore encryption is interesting, however it is not studied extensively in the paper but built on off-the-shelf tools. The paper can be made stronger by considering different encryption methods.**
>
> Thanks for your suggestion about considering more encryption methods. Two-stage decryption is our carefully designed overall encryption strategy for the memorization-interaction-based FL framework, which considers two aspects: (1) blocking the possibility of privacy theft by the server in a semi-honest setting, i.e. Content Encryption, and (2) avoiding the possibility of privacy theft by the client, designed as (K, V)-Encryption. For the specific implementation of each stage of encryption, we chose the off-the-shelf tool from the trade-off of efficiency and performance. We will explore more efficient and secure encryption methods in the future.
>
> Further, we point out that using our memory-interaction-based FL paradigm can bring additional benefits from the privacy view in contrast to the previous FL paradigm. Specifically, from the server view, the privacy protection of client information is established on the standard and lightweight RSA mechanism in our paradigm. However, in the previous FL paradigm, the construction of a privacy-protection protocol needs to introduce other expensive mechanisms such as homophonic encryption or secure multi-party computation.
>
> > **Q4: The paper does not discuss the privacy preserving ability of different approaches. For example, how do multi-round model-based interaction methods preserve the privacy of clients compared to FedNN?**
>
> Thank you for suggesting that we discuss privacy-preserving strategies for different approaches. We use FedAvg as an example to discuss how the multi-round model-based interaction-based approach preserves privacy. Di et al.$^{[1]}$ demonstrate that it is possible to obtain private information about the client from the uploaded gradient/model. Therefore in a semi-honest setting (described in section 3.4 of the original paper), we need to avoid the server obtaining any useful privacy information from the model/gradients uploaded by the client. The existing solution is similar to Content Encryption, where the client performs encryption before uploading the model to the server. Usually, differential privacy and homomorphic encryption schemes are used, which affect the final performance. In the experiments of our paper, encryption of FedAvg's interactive objects (models) was not considered, so FedAvg's performance would be further degraded in a more realistic scenario. Moreover, FT-Ensemble suffers from a similar situation as FedAvg. In contrast, the Content Encryption used by our FedNN does not impact the final performance. We would add more of these discussions and descriptions in the future version.
>
> > **Q5: Some formatting/grammar/typo.**
>
> Thank you very much for your detailed suggestions on the writing. We will improve the paper based on them.
>
> **Reference:**
> [1] Di et al. Secure federated matrix factorization. IEEE Intelligent Systems.

---

> ### Author Response · Authors · 2022-12-09
> **Re: Thanks again!**
>
> We thank the reviewer for the review. We have provided clarification in the author's response. Should the reviewer have any further suggestions, we would be happy to provide further clarification and revise our manuscript.

---

### Official Review · Reviewer_Ljh8 · 2022-10-26

**Confidence:** 3
**Correctness:** 4
**Technical Novelty And Significance:** 2
**Empirical Novelty And Significance:** 3
**Recommendation:** 6

**Clarity, Quality, Novelty And Reproducibility:**

Clarity: clear

Quality：high

Novelty: incremental

Reproducibility: low


**Strength And Weaknesses:**

Strength

In general, this paper is well organized. The method is reasonable and technically sound, the experiments are well designed and basically sufficient to support the conclusions.

Weakness

1. The innovation in the approach proposed by the authors is incremental. It is a combination of machine translation methods (e.g., kNN-MT) and federated learning methods.

2. There is too little related works about federated learning.


**Summary Of The Paper:**

The authors propose a federated nearest neighbor (FedNN) method for machine translation. Instead of multi-round model-based interactions, this mothed leverages one-round memorization-based interaction to share knowledge across different clients to build low-overhead privacy-preserving systems.

**Summary Of The Review:**

Authors propose a federated nearest neighbor (FedNN) method for machine translation. The proposed method combines federated learning and neural machine translation to achieve SOTA results. However, the innovation is limited and do not give much context to federated learning.

---

> ### Author Response · Authors · 2022-11-19
> **Response to Reviewer Ljh8**
>
> We would like to thank for your time and all nice suggestions towards our paper! We will polish the expressions and revise the paper carefully. According to your comments, we provide the responses as follows:
>
> >**Q1: The innovation in the approach proposed by the authors is incremental.**
>
> Please refer to Q1 of General Response.
>
> >**Q2: There is too little related works about federated learning.**
>
> Thank you very much for pointing out the shortcomings in related work. We will add more related work on federated learning to our paper, which is not limited to the NLP field.
>
> >**Q3: Concerns about method reproducibility.**
>
> We will open source the FedNN code and the corresponding dataset on the Github. We believe these can help the community to reproduce the work easily.

---

> ### Author Response · Authors · 2022-12-09
> **Re: Thanks again!**
>
> We thank the reviewer for the review. We have provided clarification in the author's response. Should the reviewer have any further suggestions, we would be happy to provide further clarification and revise our manuscript.

---

### Official Review · Reviewer_1QjM · 2022-10-31

**Confidence:** 4
**Correctness:** 3
**Technical Novelty And Significance:** 3
**Empirical Novelty And Significance:** 3
**Recommendation:** 6

**Clarity, Quality, Novelty And Reproducibility:**

The paper is generally well written and clear with the exception of some typos etc. It is easy to follow but I would perhaps clarify the privacy/encryption related information sooner.

There is no novel method proposed per-se, but the novelty lies in integrating the kNN MT method with federated learning and addressing the privacy and computational concerns. It is a solid contribution with nice insights, but some issues that could be improved in the experimental setup and the implementation of the methods comparing against.

In terms of reproducibility, the method proposed by the authors could probably be reproduced with some effort and using the information in the appendix, but some details for the methods compared against are less clear (see also my comments in the previous section).

**Strength And Weaknesses:**

The paper is well motivated and the authors perform comparisons across different aspects. They draw useful insights on the impact of the data distribution, the number of clients etc. They also do show that their approach outperforms the other models, especially for the non-IID setup. Overall, it is interesting to see the integration of the kNN MT approach in the federated learning workflow and the authors do a good job at addressing privacy and computational/communication concerns.

However, I have a few concerns regarding the experimental setup and decisions. The authors seem to compare against the FedAvg method which is an implementation of the McMahan et al. (2017) method, and they point out the vast communication cost. However, there have been more recent papers (that are cited by the authors as related work: Roosta et al., 2021; Passban et al., 2022) that show improvements in terms of communication. It would make more sense to compare to these methods instead, or at least make some attempt to calculate an optimal update interval for the FedAvg method instead of considering only the FedAvg_1 and FedAvg_infinity. Similarly it is unclear how the authors aggregate the prediction outputs for the FT-Ensemble method: in this case too, the aggregation approach (e.g. is it a weighted fusion?) and potential optimisation of the parameters could impact the performance of the FT-Ensemble.

Moreover, it is not clear why the authors did not use the WMT14 De-En language pair (but use the WMT14 En-De instead) that was used by the previous works and would allow for direct comparisons with reported results in other work (especially since the authors seem to have not optimised the methods they compare against).

Additionally, the authors mention significantly outperforming other methods (e.g.: "FedNN has a significant performance improvement of 4.41/1.99 BLEU scores...") but it is unclear what is the statistical significance test that was performed (if any).

While my main concerns are listed above, I would like to add some further suggestions:
- While I appreciate the already performed ablation tests, I am missing some ablation regarding the impact of the size of the data on the client and the server side. It is hinted in the paper that the size of the client data impacts the performance of the FedAvg methods but there is no specific experiment about it. Still it seems to be a crucial parameter that could impact the performance and cost of most of the compared methods and it would be very interesting to see a more detailed analysis about this.
- I would propose that the authors consider comparing the performance of different systems using some more recent, neural metric (e.g. BLEURT, COMET) in addition to BLEU as it would allow to draw better comparisons.

Some formatting/grammar/typo comments:
- Some fonts of figure 1 are way too small
- In the mention: "the server builds (K, V)-encryption model" I think it should be:  "the server builds a (K, V)-encryption model " . Additionally, you mention (K,V) a couple of times before explaining what (K,V) stands for. I would propose to add the definition/explanation in the first mention.
- "vis kNN retrieval based on the context representation" —> via instead of vis?

**Summary Of The Paper:**

The authors present an adaptation of the kNN MT model into the federated learning framework. The motivation is to achieve client-side machine translation with minimal communication costs with the central server NMT model and while maintaining the privacy of client data. The authors propose the kNN federated learning approach paired with (K,V) encryption of the data-stores as an alternative to federated learning approaches that use multi-round model updates to integrate client-side data to the central model. The authors compare their model to other baseline approaches regarding BLEU score as well as communication, computational and inference costs and they experiment with IID and non-IID setups, while they also perform ablation tests regarding the number of clients and the heterogeneity degree of data.

**Summary Of The Review:**

Well-motivated approach with interesting insights, but some questionable choices in the experimental setup that could be improved or further clarified/justified.

---

> ### Author Response · Authors · 2022-11-19
> **Response to Reviewer 1QjM (Part 1)**
>
> We would like to thank for your time and all nice suggestions towards our paper. We will polish the expressions and revise the paper carefully.
>
> > **Q1: Compare FedNN with recent solutions that have been improved for FedAvg's huge communication cost.**
>
> We compare the performance (BLEU) and overhead of FedNN, FedAvg, and Controller in the Non-IID setting of the En2De translation task. For the Controller model, as shown in Roosta's$^{[1]}$ study, 6E-6D/C-C(0-3) model achieves the best trade-off of performance and efficiency among all FL methods. Thus, we follow their setup and adopt layers 0 and 3 (both for the encoder and decoder) as controllers to participate in the parameter interaction of FedAvg training. The experimental results are shown in the table below. We find that the Controller has a significant performance improvement compared to the Public Model, but still, gaps compared to FedAvg and our FedNN. In addition, since the Controller is still a multi-round model-based FL interaction paradigm, its communication overhead is still much higher than our FedNN. Due to the short response period, we could not reproduce more methods (i.e., Dynamic Pulling$^{[2]}$ and 8E-8D$^{[1]}$) in time, which we will add to the paper in the future version.
>
> | En2De **Model**        | **IT** | **Law** | **Medical** | **WMT** | **Client Avg** | **△** | **Global Avg** | **△** | Comm. (GB) |      |
> | ---------------------- | :----: | :-----: | :---------: | :-----: | :----------: | :---- | :---------: | :---- | :--------: | ---- |
> | Public Model           | 26.62  |  35.91  |    30.27    |  **26.63**  |    30.93     | -     |    29.86    | -     |     -      |      |
> | Controller (6E-6D,0-3) | 27.78  |  46.30  |    35.62    |  18.72  |    36.57     | +5.63  |    32.11    | +2.25  |   10.86    |      |
> | FedAvg                 | 28.26  |  53.00  |    45.90    |  13.45  |    42.39     | +11.45 |    35.15    | +5.30  |   388.12   |      |
> | FedNN                  | **35.62**  |  **55.57**  |    **49.21**    |  22.29  |    **46.80**     | **+15.87** |    **40.67**    | **+10.82** |    **5.08**    |      |
>
> > **Q2: Make some attempts to calculate an optimal update interval for the FedAvg method.**
>
> Thank you for your suggestions on the FedAvg ablation study. We actually have conducted experiments to verify the impact of model interaction frequency on the final performance, but forget to add this content in the paper. We will append these results in the future version. For the experiment setting, we assume that each client uploads the model to the server for aggregation after $k\in \{1,2,5,10,20,\infty\}$ rounds of local updates. These experimental results (BLEU) are shown in the following table. We find that the final model's performance decreases significantly as k increases, especially when $k=\infty$ (i.e., a copy of the server model is trained locally until convergence), and the average performance drops to a level similar to that of PublicModel. This is because too many local updates can lead to catastrophic forgetting of knowledge from the previous aggregated models, which leads to strong knowledge conflicts in the new round of interactions. The optimal performance is presented at k=1, which means that for FedAvg, frequent interactions are essential to alleviate the knowledge conflicts frequently.
>
> | | En2De Model         | **IT**    | **Law**   | **Medical** | **WMT**   | **Client Avg** | **△**      | **Global Avg** | **△**     |
> | ------------------- | --------- | :---------: | :-----------: | :---------: | :------------: | :----------: | :-----------: | :---------: | :---------: |
> | | **Public Model**    | 26.62     | 35.91     | 30.27       | 26.63     | 30.93        | -          | 29.86       | -         |
> | **FedAvg**| **k=1**      | **28.26** | **53.00** | **45.90**   | **13.45** | **42.39**    | **+11.45** | **35.15**   | **+5.30** |
> | | **k=2**      | 26.37     | 52.92     | 44.22       | 13.07     | 41.17        | +10.24     | 34.15       | +4.29     |
> | | **k=5**      | 24.93     | 52.23     | 41.08       | 12.85     | 39.41        | +8.48      | 32.77       | +2.92     |
> | | **k=10**   | 22.63     | 51.11     | 38.60       | 12.65     | 37.45        | +6.51      | 31.25       | +1.39     |
> | | **k=20**   | 21.82     | 49.53     | 36.24       | 12.75     | 35.86        | +4.93      | 30.09       | +0.23     |
> | | **k=$\infty$** | 17.30     | 47.06     | 30.61       | 13.33     | 31.57        | +0.63      | 27.01       | -2.85     |
>
> **Reference:**
> [1] Roosta et al. Communication-efficient Federated Learning for Neural Machine Translation. arXiv:2112.06135.
> [2] Passban et al. Training Mixed-Domain Translation Models via Federated Learning. NAACL2022.

---

> ### Author Response · Authors · 2022-11-19
> **Response to Reviewer 1QjM (Part 2)**
>
> > **Q3: How to aggregate the predicted output of FT-Ensemble methods?**
>
> In fact, the ensemble method can be implemented in two ways: the first is to directly average the probability distribution of each client model (as used in the original Table I), i.e., FT-Ensemble; the second, similar to FedAvg, is to weight the probability distribution of each client model's output by assigning weights to it according to its data size, i.e., FT-Ensemble-Wei. The performance  (BLEU) comparison of these two ways in the Non-IID setting is shown in the following table. We find that FT-Ensemble outperforms FT-Ensemble-Wei in both client-side and global overall performance. FT-Ensemble has a more balanced performance on the client side, while FT-Ensemble-Wei is similar to FedAvg in that the performance is more biased towards the client Law's model, which has more data volume. Our FedNN outperforms both of these methods on all test sets. Note that the two implementations of FT-Ensemble described above in the IID setup are equivalent since the data size is the same for each client. The IID result is shown in Table 1 of the original paper.
>
> | **En2De Non-IID Model** | **IT**    | **Law**   | **Medical** | **WMT**   | **Client Avg** | **△**     | **Global Avg** | **△**     |
> | ----------------------- | :---------: | :---------: | :-----------: | :---------: | :------------: | :---------: | :-----------: | :---------: |
> | **Public Model**            | 26.62     | 35.91     | 30.27       | **26.63** | 30.93        | -         | 29.86       | -         |
> |**FT-Ensemble**        | 30.11     | 38.14     | 39.15       | 17.13     | 35.80        | 4.87      | 31.13       | 1.28      |
> | **FT-Ensemble-Wei**         | 24.09     | 48.58     | 34.11       | 16.06     | 35.59        | 4.66      | 30.71       | 0.85      |
> | **FedNN**                   | **35.62** | **55.57** | **49.21**   | 22.29     | **46.80**    | **15.87** | **40.67**   | **10.82** |
>
> > **Q4: Performance comparison in the De->En translation direction.**
>
> The following table compares each method's performance (BLEU) in the Non-IID setting. We observe that the findings in the De->En direction remain consistent with the En->De direction, in which FedNN outperforms the other methods in terms of overall performance both client-side and globally. Due to the short Response period and resource constraints, experimental results on the IID setting will be added to the paper in the future.
>
> | **Model**        | **IT**    | **Law**   | **Medical** | **WMT**   | **Client Avg** | **△**      | **Global Avg** | **△**      |
> | ---------------- | --------- | --------- | ----------- | --------- | ------------ | ---------- | ----------- | ---------- |
> | **Public Model** | 31.70     | 39.86     | 34.37       | **31.64** | 35.31        | -          | 34.39       | -          |
> | **FedAvg$_1$**   | 32.22     | 58.32     | 48.56       | 16.83     | 46.37        | +11.06     | 38.98       | +4.59      |
> | **FT-Ensemble**  | 35.76     | 44.07     | 43.20       | 21.48     | 41.01        | +5.70      | 36.13       | +1.74      |
> | **FedNN**        | **41.11** | **60.18** | **53.44**   | 27.12     | **51.58**    | +**16.27** | **45.46**   | +**11.07** |
>
> > **Q5: Add the results of significance tests between FedNN and other methods.**
>
> We use the bootstrap re-sampling$^{[1]}$ (Koehn, 2004) method to test the significant difference between FedNN and other methods. The following table shows the significance test results of En->De under the Non-IID setting. The "-" means that FedNN is not significantly better than the method. We can find that FedNN outperforms all existing FL methods, including FedAvg and FT-Ensemble.
>
> | **Model**        |  **IT**   |  **Law**  | **Medical** | **WMT14** |
> | ---------------- | :-------: | :-------: | :---------: | :---------: |
> | **Global Model** |     -     | $\le$0.01 |  $\le$0.05  | -  |
> | **Public Model** | $\le$0.01 | $\le$0.01 |  $\le$0.01  | $\le$0.01  |
> | **FedAvg**       | $\le$0.01 | $\le$0.01 |  $\le$0.05  | $\le$0.01  |
> | **FT-Ensemble**  | $\le$0.01 | $\le$0.01 |  $\le$0.01  | $\le$0.05  |
>
> **Reference:**
>
> [1] Philipp Koehn. Statistical Significance Tests for Machine Translation Evaluation. EMNLP2004.

---

> ### Author Response · Authors · 2022-11-19
> **Response to Reviewer 1QjM (Part3)**
>
> >  **Q6: missing some ablation regarding the impact of the size of the data on the client side.**
>
> Thank you for your insightful suggestions. We perform this ablation experiment in the Non-IID setting of En->De direction. Specifically, for each domain, we adopt a ratio range of $\beta\in\{0.0, 0.2,0.4,0.6,0.8,1.0\}$ to randomly sample from its complete data to constitute the data at the client side of different scales for quick experiments. Experimental results are shown in the following table. We find that the performance and cost of all methods increase as the amount of data on the client-side increases. In addition, our FedNN outperforms other FL methods, including FedAvg and FT-Ensemble, regardless of the proportional size of the data scale. Due to the short Response period and resource constraints, experimental results on the IID setting will be added to the paper in the future.
>
> |                   |              | IT    | Law   | Medical | WMT   | Client Avg | △     | Global Avg | △     | Comm. (GB) |
> | :---------------: | :----------- | :---: | :---: | :-----: | :---: | :------: | :---- | :-----: | :---- | :--------- |
> | $\beta=0.0$ | **Public Model** | 26.62 | 35.91 | 30.27   | 26.63 | 30.93    | -     | 29.86   | -     | -          |
> | $\beta=0.2$ | **FedAvg$_1$** | 20.27 | 47.42 | 33.92   | 15.73 | 33.87    | +2.94 | 29.34   | -0.52 | 111.59     |
> |  | **FT-Ensemble**   | **29.95**        | 40.25 | 37.41 | 20.81   | 35.87 | +4.94    | 32.11 | +2.25   | 4.85  |
> |  | **FedNN**         | 29.65        | **46.89** | **41.46** | **22.72**   | **39.33** | **+8.40**    | **35.18** | **+5.32**   | **2.00**  |
> | $\beta=0.4$ | **FedAvg$_1$** | 20.99 | 48.98 | 35.58   | 16.58 | 35.18    | +4.25 | 30.53   | +0.67 | 116.44     |
> |  | **FT-Ensemble**   | 29.60        | 39.50 | 38.45 | 21.43   | 35.85 | +4.92    | 32.25 | +2.39   | 4.85  |
> |  | **FedNN**         | **30.41**    | **50.30** | **43.89** | **23.32** | **41.53** | **+10.60** | **36.98** | **+7.12** | **2.77** |
> | $\beta=0.6$ | **FedAvg$_1$** | 22.36 | 52.38 | 42.04   | 12.83 | 38.93    | +7.99 | 32.40   | +2.55 | 245.00     |
> |  | **FT-Ensemble**   | 28.32        | 40.55 | 40.19 | 18.61   | 36.35 | +2.48    | 31.92 | +2.06   | 4.85  |
> |  | **FedNN**         | **31.65**    | **52.29** | **45.65** | **23.68** | **43.20** | **+12.26** | **38.32** | **+8.46** | **3.54** |
> | $\beta=0.8$ | **FedAvg$_1$** | 24.11 | 52.71 | 43.72   | 12.76 | 40.18    | +9.25 | 33.33   | +3.47 | 354.74     |
> |  | **FT-Ensemble**   | 29.01        | 38.16 | 39.30 | 16.87   | 35.49 | +4.56    | 30.84 | +0.98   | 4.85  |
> |  | **FedNN**         | **32.15**    | **54.11** | **46.62** | **22.22** | **44.29** | **+13.36** | **38.78** | **+8.92** | **4.31** |
> | $\beta=1.0$ | **FedAvg$_1$** | 28.26 | 53.00 | 45.90   | 13.45 | 42.39    | +11.45 | 35.15   | +5.30 | 388.12     |
> |  | **FT-Ensemble**   | 30.11        | 38.14 | 39.15 | 17.13   | 35.80 | +4.87    | 31.13 | +1.28   | **4.85** |
> |  | **FedNN**         | **35.62**    | **55.57** | **49.21** | **22.29** | **46.80** | **+15.87** | **40.67** | **+10.82** | 5.08  |

---

> ### Author Response · Authors · 2022-11-19
> **Response to Reviewer 1QjM (Part 4)**
>
> > **Q7: Consider comparing the performance of different methods using neural metric BLEURT.**
>
> Thanks for your suggestions on evaluation metric!  We evaluate the two settings in Table 1 of the original paper using the neural metric - BLEURT. Experimental results are shown in the table below. We can obtain similar conclusions when using the BLEU score as an evaluation metric, i.e., in the Non-IID setting, our FedNN significantly outperforms all other FL methods; in the IID setting, our FedNN also achieves comparable performance to the  FedAvg$_1$ and FT-Ensemble.
>
> |         | Model               |    IT     |    Law    |  Medical  |    WMT    | Client Avg |         △ | Global Avg |         △ |
> | ------- | ------------------- | :-------: | :-------: | :-------: | :-------: | :--------: | --------: | :--------: | --------: |
> |         | Centralized Model   |   70.46   |   78.49   |   74.97   |   71.62   |   74.64    |         - |   73.89    |         - |
> |         | Public Model        |   62.00   |   72.65   |   65.64   |   71.93   |   66.76    |         - |   68.06    |         - |
> | Non-IID | FedAvg$^s_1$        |   63.02   |   73.71   |   67.06   |   72.19   |   67.93    |     +1.17 |   69.00    |     +0.94 |
> |         | FedAvg$_1$          |   64.09   |   78.05   | **72.74** |   53.61   |   71.63    |     +4.86 |   67.12    |     -0.93 |
> |         | FedAvg$^s_{\infty}$ |   62.87   |   74.11   |   67.50   | **72.20** |   68.16    |     +1.40 |   69.17    |     +1.11 |
> |         | FedAvg$_{\infty}$   |   52.63   |   77.37   |   65.02   |   56.26   |   65.01    |     -1.76 |   62.82    |     -5.24 |
> |         | FT-Ensemble         |   64.08   |   72.47   |   70.01   |   59.17   |   68.85    |     +2.09 |   66.43    |     -1.62 |
> |         | FedNN               | **68.86** | **78.12** | **72.74** |   67.38   | **73.24**  | **+6.48** | **71.78**  | **+3.72** |
> | IID     | FedAvg$^s_1$        |   66.38   |   76.31   |   71.65   | **72.46** |   71.45    |     +4.68 |   71.70    |     +3.65 |
> |         | FedAvg$_1$          |   69.93   |   78.67   | **75.68** |   55.78   | **74.76**  | **+8.00** |   70.02    |     +1.96 |
> |         | FedAvg$^s_{\infty}$ |   64.92   |   74.32   |   68.68   |   72.15   |   69.31    |     +2.54 |   70.02    |     +1.96 |
> |         | FedAvg$_{\infty}$   |   68.96   |   78.08   |   74.39   |   59.44   |   73.81    |     +7.05 |   70.22    |     +2.16 |
> |         | FT-Ensemble         | **70.28** | **78.80** |   75.04   |   58.78   |   74.71    |     +7.94 |   70.73    |     +2.67 |
> |         | FedNN               |   67.69   |   77.74   |   72.31   |   68.16   |   72.58    |     +5.82 | **71.48**  | **+3.42** |
>
> > **Q8: Some formatting/grammar/typo.**
>
> Thank you very much for your detailed suggestions on the writing. We will improve the paper based on them.

---

> ### Author Response · Authors · 2022-12-09
> **Re: Thanks again!**
>
> We thank the reviewer for the review. We have provided clarification in the author's response. Should the reviewer have any further suggestions, we would be happy to provide further clarification and revise our manuscript.

---

### Author Response · Authors · 2022-11-19
**General Response**

We would like to thank all reviewers very much for the detailed feedback and valuable suggestions! We will follow the suggestions on experiments and writing and revise accordingly.

> **Q1:  A discussion on the novelty of the proposed method.**

In response to reviewers' concerns about the novelty of our proposed method, we provide a discussion here. Actually, the novelty of this paper is reflected in both paradigm and technical aspects.

For the paradigm aspect, we point out that the conventional multi-round model-based interaction paradigm (e.g., FedAvg) is not suitable for real-world NMT applications since it incurs huge communication/computation overhead and asynchronous waiting time. To address this problem, we explore a novel federated paradigm that requires only one-round memorization interaction. Although our paradigm is based on an existing KNN augmentation technique, the key idea of combining this technique with the FL setting has not been explored and empirically validated before. Our experiments prove the feasibility and practicality of memorization-based interaction FL workflow. In fact, it is also interesting to further explore the combination of other memorization-augmented methods, such as translation memory methods, to perform the above component. In addition, applying the KNN technique to FL settings requires designing the privacy-enhancing mechanism carefully, which is discussed in the technical part (see next paragraph).

For the technical aspect,  we provide a scheme for the memorization-interaction FL framework regarding privacy protection (two-stage encryption) and privacy assessment (threat model and corresponding metrics), which lays the cornerstone for further work to follow. Further, we point out that using our memorization-interaction-based FL paradigm can bring additional benefits from the privacy view in contrast to the previous FL paradigm. Specifically, from the server view, the privacy protection of client information is established on the standard and lightweight RSA mechanism in our paradigm. However, in the previous FL paradigm, the construction of a privacy-protection protocol needs to introduce other expensive mechanisms, such as homophonic encryption or secure multi-party computation.

Taken together, we hope that FedNN can provide the FL community with a new perspective on performing FL workflows to avoid costly multiple rounds of model-based interactions. This work can be seen as an initial attempt to innovate the conventional FL paradigm by leveraging the recent memorization-augmentation technique. We also believe that our work can further brings insights into how to design an efficient and effective FL paradigm for those scenarios with big NN models to the research community.  We would add more of these discussions and descriptions in the future version. We would also open source the FedNN code and the corresponding dataset on the Github. We believe these can help the community to reproduce the work easily.

---

### Decision · Program_Chairs · 2023-01-20

**Decision:**

Accept: poster

**Justification For Why Not Higher Score:**

I don't think the paper presents enough novelty for being accepted as spotlight or oral.

**Justification For Why Not Lower Score:**

Despite the lack of technical novelty, the paper puts two lines of research together and it does it well. Experiments are convincing and include ablation tests.

**Metareview: Summary, Strengths And Weaknesses:**

This paper puts together retrieval-augmented machine translation (via the kNN-MT) and federated learning to achieve privacy-preserving client-side machine translation with minimal communication costs with the central server NMT model. The authors compare their model to other baseline approaches regarding BLEU score as well as communication, computational and inference costs and perform several ablation tests. In their updated version, the authors report also BLEURT scores (trained neural metrics) which correlate better with human judgments than BLEU. This is a somewhat a borderline paper, with all reviewers inclined to accept, but pointing out the limited technical novelty as a concern, as the paper essentially puts together two lines of research, kNN-MT and federated learning. In my opinion, despite the lack of technical novelty, putting these two things together is still a valuable contribution and the result is of interest to the research community, therefore I lean towards acceptance. I urge the authors to incorporate the suggestions made by the reviewers when preparing the final version of their paper.

**Note From Pc:**

if the above contains the word "oral" or "spotlight" please see: "oral" presentation means -> notable-top-5% and "spotlight" means -> notable-top-25%. As stated in our emails, we are disassociating presentation type from AC recommendations

**Summary Of Ac-Reviewer Meeting:**

Unfortunately there was no meeting. I tried to schedule a meeting (twice) but the reviewers were not responsive. Only one reviewer answered my quest for discussion and her points were helpful for my final recommendation.